# Membrane shape-mediated wave propagation of cortical protein dynamics

Zhanghan Wu[1], Maohan Su[2], Cheesan Tong[2], Min Wu [2] & Jian Liu[1]

Immune cells exhibit stimulation-dependent traveling waves in the cortex, much faster than typical cortical actin waves. These waves reflect rhythmic assembly of both actin machinery and peripheral membrane proteins such as F-BAR domain-containing proteins. Combining theory and experiments, we develop a mechanochemical feedback model involving membrane shape changes and F-BAR proteins that render the cortex an interesting dynamical system. We show that such cortical dynamics manifests itself as ultrafast traveling waves of cortical proteins, in which the curvature sensitivity-driven feedback always constrains protein lateral diffusion in wave propagation. The resulting protein wave propagation mainly reflects the spatial gradient in the timing of local protein recruitment from cytoplasm. We provide evidence that membrane undulations accompany these protein waves and potentiate their propagation. Therefore, membrane shape change and protein curvature sensitivity may have underappreciated roles in setting high-speed cortical signal transduction rhythms.

[1] Biochemistry and Biophysics Center, National Heart, Lung, and Blood Institute, National Institutes of Health, Bethesda, MD 20892, USA. [2] Department of Biological Sciences, Centre for Bioimaging Sciences, Mechanobiology Institute, National University of Singapore, Singapore 117557, Singapore. Zhanghan Wu and Maohan Su contributed equally to this work. Correspondence and requests for materials should be addressed to M.W. (email: dbswum@nus.edu.sg) or to J.L. (email: jian.liu@nih.gov)

Propagating protein waves have been observed in many cell types, sometimes associated with signal transduction[1]. Such waves on plasma membrane are especially intriguing, and underlie important cellular functions[2]. For instance, neutrophils use self-organizing features of actin traveling waves to control cell motility and avoid barriers during chemotaxis[3]. Many models have been developed to account for diverse cortical actin waves[2]. Most commonly in these models, the plasma membrane is treated as a passive and flat two-dimensional manifold; consequently, these waves are modeled essentially as chemical waves that arise from the excitability of cortical reaction-diffusion processes.

Our recent findings indicated that membrane shape might play an important role for cortical rhythmic propagation[4]: upon antigen stimulation in immune cells, not only the actin machinery (e.g., actin, N-WASP, and Cdc42), but also F-BAR domain-containing proteins—well known for their sensitivities for membrane curvature[5–7]—exhibit traveling wave behavior within the ventral cortex. Remarkably, these waves travel as fast as ~1 µm s$^{-1}$, about 10- to 100-fold higher than previously described ventral actin waves[3,8–16]. This fast wave speed together with the involvement of F-BAR proteins raise the questions regarding the underlying physical mechanism and the role of the membrane for these traveling waves.

It has been recognized that membrane shape change can feed back onto the biochemical reaction kinetics governing many cellular processes[17–21]. Considering that F-BAR proteins are members of BAR superfamily proteins that can sense and generate membrane curvature[5–7,22–28], we reasoned that these cortical proteins might mediate membrane shape changes that are important to the wave propagation. We combine theory and experiment to test this hypothesis. We show that feedback between membrane shape changes and recruitment of curvature sensing proteins can lead to traveling waves, in which propagating membrane undulation accompanies the cortical protein waves. Such curvature sensitivity-driven feedback not only constrains protein lateral diffusion in wave propagation but also enables wave propagation that outruns protein diffusional limit along the membrane. Curvature sensing thus may be important in high-speed cortical signal transmission.

## Results

**Model development.** Our model is based on the notion that membrane shape feeds back with cortical protein recruitment[19,21,29,30]. Such feedback can generate wave behavior[30], and similar ideas have been proposed to account for traveling waves in other systems[29,31]. Here we set out to explore how such concept plays out in our own system. We aimed to construct a minimal model—including the most essential components as demonstrated by existing experiments[4]—to recapitulate the basic phenomena of our traveling waves (Fig. 1a). To build the model, we extensively analyzed literature data on the physicochemical properties of, and the interactions between, F-BAR proteins, actin machinery, and membrane. This analysis allows us to obtain quantitative estimates for a range of relevant model parameters.

Our previous experiments showed that multivalent antigen crosslinking of Fc-receptors on the cell surface locally and transiently activates Cdc42[4]. FBP17 is an effector of Cdc42 and binds to active GTP-bound Cdc42 through its HR1 domain. As observed by total internal reflection fluorescence microscopy (TIRFM), waves of active Cdc42 were synchronized with waves of FBP17, and preceded actin waves[4]. Based on these experiments, the model assumed a transient activation signal that increases the local concentration of GTP-bound Cdc42 at the cortex (Fig. 1a). A similar activation signal may be evoked by the intrinsic cellular state via a self-assembly process that does not require external

stimulation, as in the case of spontaneous traveling waves[4]. Regardless of its origin, GTP-Cdc42 on the membrane was assumed to recruit effector proteins, including N-WASP and F-BAR domain-containing proteins such as FBP17 and CIP4[4,32,33], which we modeled as a generic entity designated "F-BAR".

The resulting cortical Cdc42, N-WASP, and F-BAR, which we called the "CWF" module, was assumed to have two effects. First, F-BAR deforms the membrane shape[23,24,26,27,34]. Here we focused on the low cortical density limit, where F-BAR causes a mild membrane shape deformation, rather than the sharply curved membrane tubulation seen in the high-density limit[7]. As curvature sensing is most effective in the low density limit[5–7], we postulated that the resulting membrane shape deformation induced further F-BAR recruitment to the cortex. The additional cortical F-BAR would then bind more Cdc42 and N-WASP. Interplay among Cdc42, N-WASP, and F-BAR thus positively feeds back with the membrane shape changes. Second, based on experimental observations[35–37], the model assumed that the active N-WASP promoted branched actin polymerization via Arp2/3 complex activation. Although actin polymerization was required for wave nucleation[4], actin polymerization also antagonized F-BAR cortical recruitment in at least two ways. Cooperation between actin polymerization and dynamin could trigger membrane fission[26], leading to F-BAR disassembly. Alternatively, actin polymerization could stiffen the cortex via increased membrane-actin cytoskeleton coupling[38–40], which makes the membrane less deformable, attenuating the membrane shape-mediated positive feedback with F-BAR recruitment. We assumed cortex stiffening by actin polymerization in our model, although other mechanisms would have similar effects. Therefore, increasing actin polymerization in our model enhanced local membrane/cortex tension that negatively fed back to cortical recruitment of CWF.

The essence of the model is contained in a set of partial differential equations (Eqs. (1.1)–(1.8) in Supplementary Note 1) that govern the membrane shape changes and reaction-diffusion processes of the associated cortical proteins. The model describes cortical protein recruitment from and turnover into the cytoplasm by Michaelis–Menten-like kinetics. For simplification, the cytoplasm was treated as an unlimited reservoir. Once recruited to the membrane, proteins diffused laterally, although more slowly than their cytoplasmic counterparts[41,42]. We modeled the membrane as an elastic sheet, whose shape was governed by membrane tension, bending modulus, membrane-substrate adhesion, and mechanical actions of cortical proteins. These mechanical actions included actin-mediated enhancement of membrane tension and F-BAR-mediated membrane shape deformation. Finally, the resulting membrane shape altered the rate of F-BAR cortical recruitment, a.k.a. curvature sensing. Thus, the model described mechanochemical feedback between membrane shape and protein curvature sensitivity.

We simulated the model on a membrane patch of 40 µm diameter that represents the ventral cortex. The model clamped the membrane patch at the edge, which recapitulated apparent hindering effect of the cell leading edge, as the traveling waves in our system were only observed on the cell ventral side. We computed the dynamical evolution of the system by integrating the equations from an initial condition—a flat membrane patch with no cortical proteins. The model output at each time step is as follows: (1) the local membrane height; and (2) the cortical protein densities across the simulation domain. To jump-start the dynamics, the model input was a localized transient GTP-Cdc42 pulse as suggested by our previous experiments[4]. Below, we present typical model results, the essence of which holds over a broad model-parameter space (see Supplementary Fig. 1 for phase diagram studies).

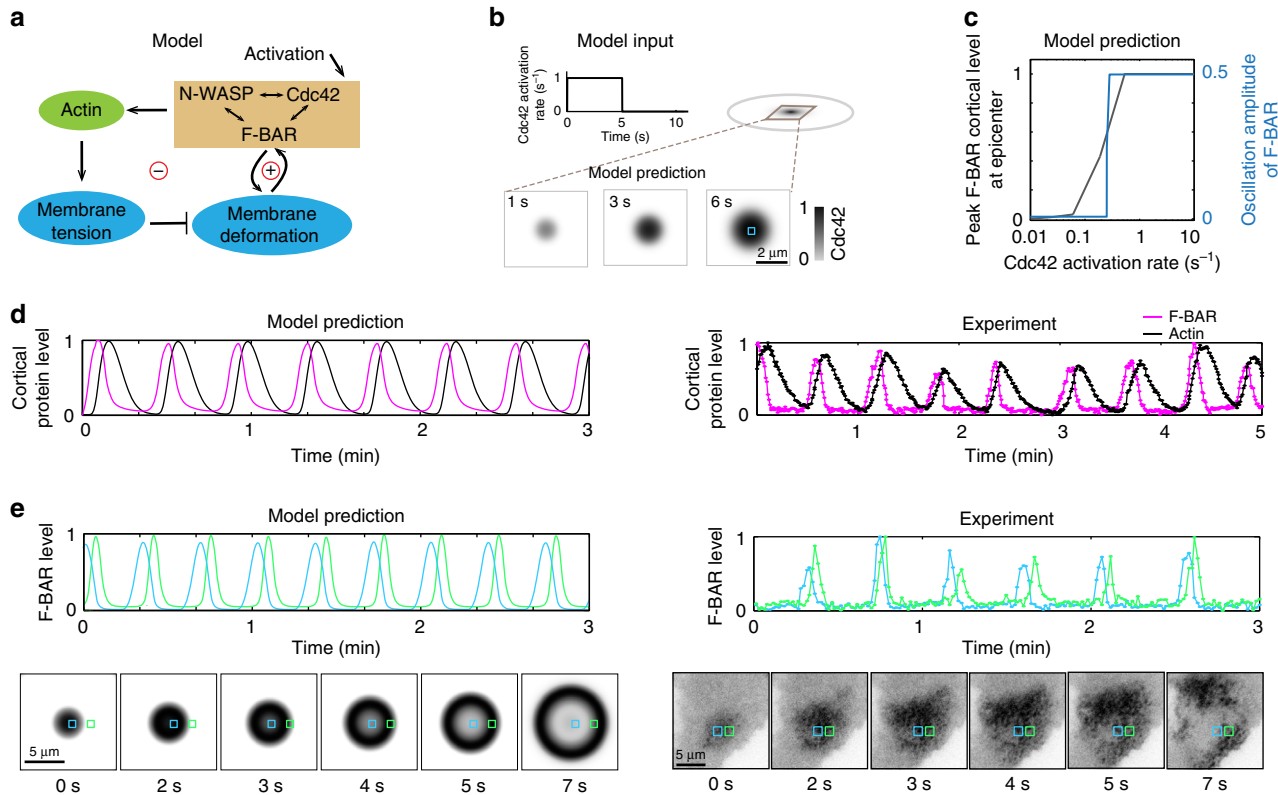

**Fig. 1** Cortical protein oscillation underlies wave initiation and propagation. **a** Key model components and their interactions. **b** Model prediction that a localized GTP-Cdc42 pulse results in nucleation of a cortical patch. In the simulation shown, this external activation was transient, i.e., after 5 s it was turned off for the rest of the simulation. Top: temporal and spatial profiles of the initial Cdc42 activation. Bottom: spatial-temporal evolution of Cdc42 cortical density. **c** Model prediction that oscillation of cortical protein requires a threshold activation level. **d** Local cortical protein oscillation during wave propagation. Left: the model prediction of the time curves of F-BAR and actin oscillation at the epicenter, indicated by a cyan box in Fig. 1b, bottom. Right: experimental results show similar oscillations of cortical proteins. **e** Characteristics of wave propagation. Left: model prediction. Right: experimental measurement. Top: stable phase shift in the cortical oscillation between neighboring areas. The locations are marked by blue squares and green squares in the corresponding cases, in which they are 2.5 μm apart in model result (left bottom) and 2.88 μm apart in experiment (right bottom). The two ROIs are chosen to show that oscillations at the different locations have a stable phase shift, a feature that holds for any two ROIs along the direction of wave propagation. Bottom: snapshots of F-BAR cortical density profiles over time. For the model result, the time is relative to when the transient Cdc42 activation stops in Fig. 1b. In the experiment, the F-BAR level is the FBP17 fluorescence intensity minus background fluorescence, normalized by the maximum fluorescence

**Wave formation**. With the model, an initial transient pulse of localized Cdc42 activation (Fig. 1b, top) evolved into a coherent and sustained Cdc42 patch with a spatial protein gradient that tapered off from the epicenter (the snapshots in Fig. 1b, bottom). Our model predicts that oscillations emerge when F-BAR concentration exceeds a threshold during nucleation process of the cortical protein patch (Fig. 1c). According to the model, CWF was the first module recruited. The activated N-WASP promoted actin polymerization by activating an Arp2/3 complex-mediated pathway. The resulting actin polymerization stiffened the cortex; this stiffening inhibited F-BAR-mediated membrane deformation and therefore antagonized further CWF formation. Due to the autocatalytic nature of the Arp2/3 complex activation, actin polymerization took some time to initiate; consequently, its inhibitory effect on CWF was delayed. A delayed negative feedback naturally gives rise to an oscillatory behavior, where peaks of CWF oscillation preceded that of actin oscillation by 6–7 s, consistent with our experimental observation (Fig. 1d).

As the epicenter began with the highest cortical protein levels (blue square, Fig. 1b, bottom), it entered oscillatory dynamics first (from $t = 2$ to 3 s, Fig. 1e). Specifically, as actin polymerization-mediated negative feedback kicked in, it decreased the CWF cortical level. This initial "collapse" at the epicenter transformed

the coherent patch into a ring-like profile with the highest intensity now at the rims (from $t = 2$ to 3 s, Fig. 1e). Subsequently, as the CWF wavefront expanded outward, the actin-mediated inhibitory effect—due to the temporal delay—kept on decreasing the CWF cortical level at the trailing side of the wavefront (from $t = 3$ to 7 s, Fig. 1e). During this expansion, the cortical areas away from the epicenter ensued oscillatory dynamics (e.g., green square in Fig. 1e). Accordingly, local oscillation at the ring periphery (green square in Fig. 1e) was with a constant phase lag relative to that at the epicenter (blue square in Fig. 1e), consistent with our experimental observation (Fig. 1e).

**Membrane shape undulation accompanies protein wave**. Critically, the model predicts that membrane shape deformation accompanies our protein traveling wave over time and space (Fig. 2a–c). Due to the curvature sensitivities of F-BAR, the membrane shape itself acts as an additional activation signal, which recruits cortical proteins that reciprocally deform the membrane (Fig. 1a). That is, membrane shape changes are not only part of the feedback important for local oscillatory response but also a carrier of the rhythmic propagation (Fig. 1a).

The snapshots in Fig. 2a provide a sense of the membrane shape changes in wave propagation (see Supplementary Fig. 2 for

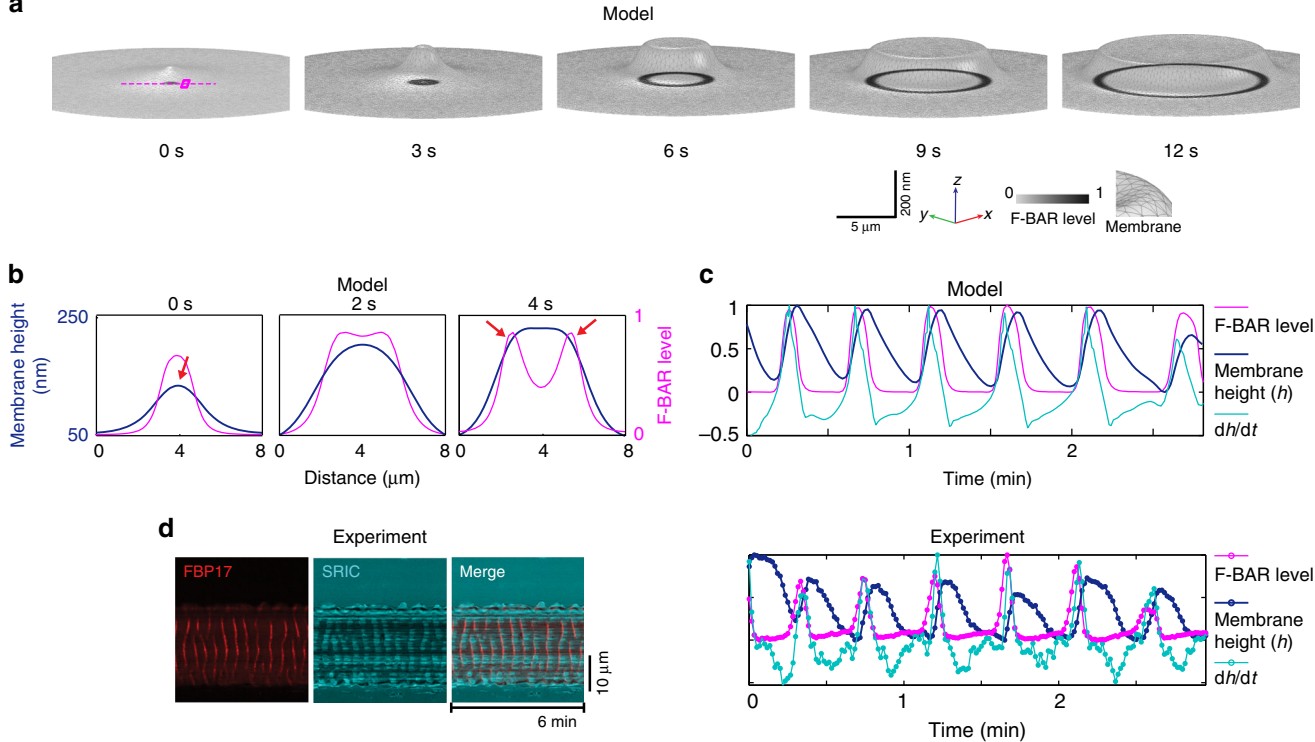

**Fig. 2** Membrane shape changes accompany traveling wave propagation. **a** Model predicts that the local membrane shape correlates with the F-BAR cortical density at the wavefront during wave propagation. The grayish meshes represent the membrane shapes in three-dimensional space (X, Y, Z) with a perspective angle; the inverted density maps are the corresponding F-BAR densities on the membrane, which is projected onto the two-dimensional X–Y plane. The scale bar for the x–y dimension is 5 µm and for the z-dimension is 200 nm. For the purpose of demonstration, the snapshots in Fig. 2a only show a part of the simulated membrane patch. The wave propagates beyond the domain defined in the snapshots until it approaches the boundary of the membrane patch; there, the wave subsides and eventually disappears, as the local membrane is clamped (Supplementary Movie 2 and Supplementary Fig. 2). **b** A zoom-in view of the local cortical rhythm along the magenta dash line across the cortex marked in Fig. 2a at $t = 0$, 2, and 4 s. Blue lines: local membrane shape. Magenta lines: F-BAR cortical level. The red arrows mark the locations with the largest membrane curvature. **c** Model prediction on temporal evolutions of the local F-BAR cortical level, membrane height, and rate of membrane height changes ($dh/dt$) at the location of the magenta square in Fig. 2a. **d** Experimental measurement of the spatial-temporal changes in F-BAR cortical level (by TIRFM) and membrane height (by SRIC) during traveling wave propagation. Left: kymographs of the local F-BAR cortical level and the corresponding relative membrane height. Right: time curves representing the local F-BAR cortical level, membrane height, and the rate of membrane height changes ($dh/dt$) at a fixed location. From SRIC measurements, we estimate the maximum membrane height to be ~140 nm (Supplementary Fig. 3). Note that this number is a very rough estimate, as the refractive indexes both inside and outside of the cell may undergo variation, due to changes in salt concentration and the organelle proximity. We therefore used the normalized membrane height in the experimental plot to avoid systematic error

a more global perspective). Figure 2b is a zoom-in view of this key feature evolving over space (along the magenta dash line in the top snapshot of Fig. 2a, also see Supplementary Movie 1). Likewise, Fig. 2c predicts the temporal correlations in the oscillations—between the membrane height and the cortical F-BAR protein level—at a fixed location marked by the magenta square in the top snapshot of Fig. 2a. To test these predictions, we simultaneously imaged FBP17 wave propagation by TIRFM and the local distance between cell membrane and the substrate by surface reflectance interference contrast (SRIC) microscopy. We found that local membrane shape deformation strongly correlated with the location of the wave marked by cortical FBP17 density (Fig. 2d, Supplementary Movie 3). Interestingly, membrane height oscillation has a phase lag relative to cortical FBP17 levels (Fig. 2d). As illustrated in Fig. 2b, this phase lag arises as a natural consequence of time delay between changes in membrane curvature and membrane height. As the wave propagates, the wavefront marks the highest F-BAR intensity and hence membrane curvature. In contrast, the membrane height continues to increase in the trailing path behind the F-BAR wavefront. This view is supported by the fact that the local cortical F-BAR density is in phase with the rate of membrane height changes ($dh/dt$) at a

fixed location, while preceding the membrane height ($h$) by a few seconds (Fig. 2c, d). We note that while SRIC is sufficient to detect significant membrane-substrate distance changes, its spatial resolution is too limited to precisely map out actual membrane curvature at finer scale[43]. Nevertheless, our findings indicate a strong correlation between local membrane shape change and rhythmic propagation. Furthermore, this observation demonstrates that F-BAR dynamics synchronizes with membrane shape deformation, suggesting curvature sensing is at play, consistent with our model proposal.

**F-BAR curvature preference is critical for wave propagation.** As membrane shape did change detectably along the wave propagation path, we determined whether membrane shape changes were necessary for wave propagation. The model predicts that wave propagation has specific requirements on the curvature sensitivity and recruitment rate of F-BAR (Fig. 3a). To test this prediction, we first examined the effect of removing F-BAR proteins on the formation of traveling waves. We reduced the level of endogenous curvature-generating proteins by shRNA. Knocking down FBP17 or its homolog CIP4 alone did not significantly reduce the percentage of the cells with traveling waves,

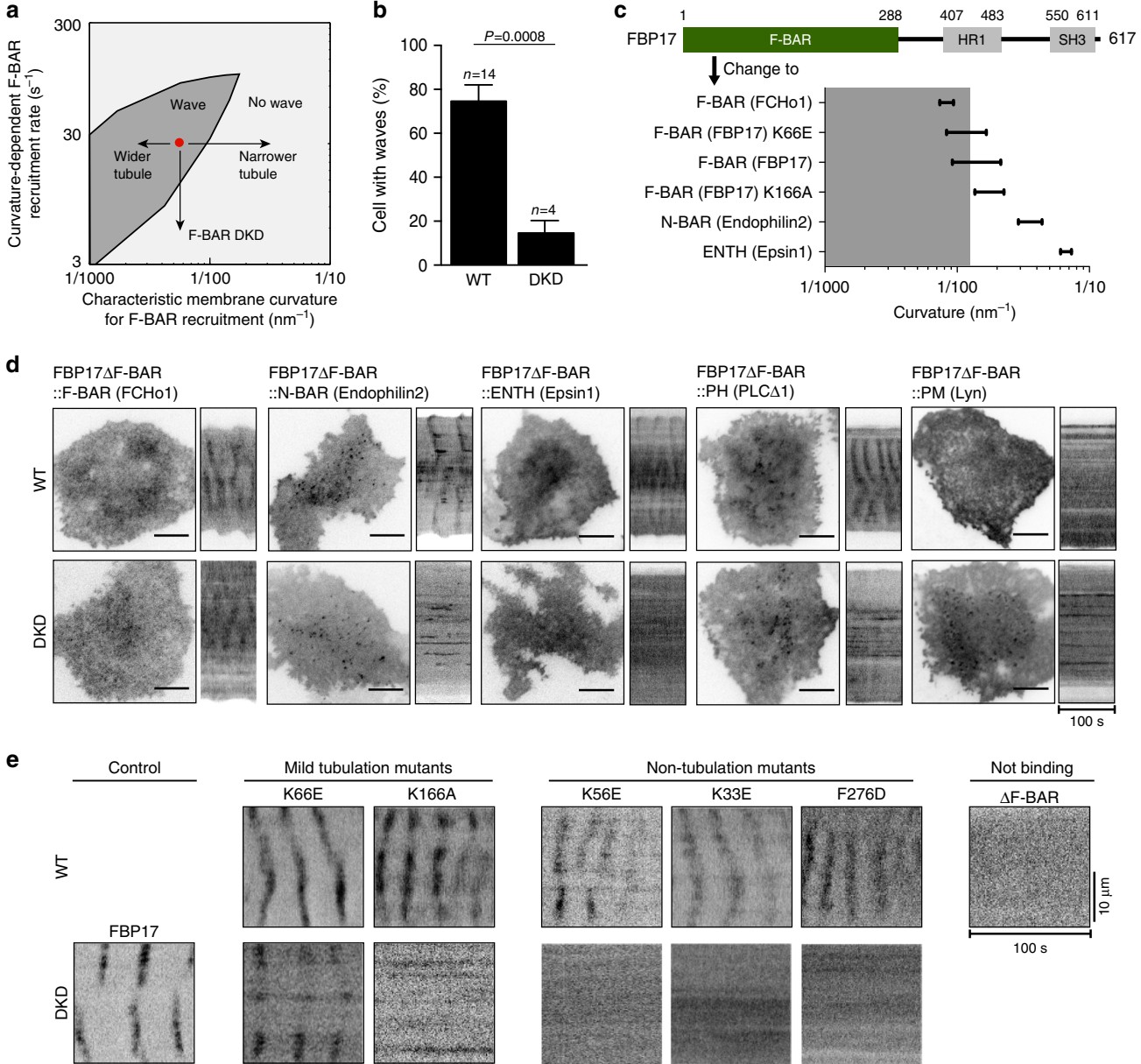

**Fig. 3** Wave propagation requires F-BAR and its curvature sensitivity. **a** Model phase diagram showing the dependence of traveling waves on a proper balance between the curvature-dependent F-BAR-recruitment rate and the characteristic membrane curvature that recruits F-BAR. The red dot represents the model parameter set that generated the nominal model results in Figs. 1 and 2. **b** Double knockdown (DKD) of FBP17 and CIP4 by shRNA causes a reduction ($P = 0.0008$, Student's $t$-test) in the percentage of cells with active Cdc42 (CBD-GFP) waves (4 experiments with 39 cells in total), comparing to wild-type (WT) cells (14 experiments with 133 cells). Error bars: s.e.m. **c** F-BAR domain of FBP17 is replaced with other membrane-binding motifs. Ranges of curvature (radius$^{-1}$) preferences of different membrane-binding domains are plotted[24, 25, 44, 45]. Gray area indicates the curvature range that allows wave formation, corresponding to the gray area of **a**. **d** The curvature of F-BAR domain is critical for wave formation. A micrograph and a kymograph for each mutant protein both in WT and DKD conditions are shown. Only the mutant protein with an F-BAR (FCHo1) domain could rescue wave formation in DKD cells. **e** The ability of point mutants of F-BAR domain of FBP17 to localize to the waves in WT cells or rescue wave formation in DKD cells correlates with their tubulation activities in vitro. Representative kymographs of cells expressing point mutants or full-length construct of FBP17-GFP in WT cells and DKD cells are shown. Mutants are separated into four groups based on curvature-generating abilities. Scale bar: 10 μm

but double knocking down (DKD) FBP17 and CIP4 abolished the formation of traveling waves (Fig. 3b). This indicates some redundancies among the Toca family of F-BAR proteins and demonstrates their essential role as a collective entity of the Cdc42 interacting F-BAR proteins in the traveling waves.

To determine whether the F-BAR domain of FBP17 was necessary or sufficient for the FBP17 wave, we generated multiple versions of truncated FBP17. Deletion of any of the three

structural domains (F-BAR, HR1, or SH3) abolished the wave-like appearance of cortical FBP17 in wild-type cells (Supplementary Fig. 4), indicating that the F-BAR domain was necessary, but not sufficient for generating the traveling waves. ΔSH3 and ΔHR1 mutants each formed some stable puncta on cell membrane, but ΔF-BAR did not (Supplementary Fig. 4). Because the SH3 domain and HR1 domain interact with actin machinery, the formation of stable FBP17 puncta in ΔSH3 and ΔHR1

mutants suggests the requirement of a link to actin for dissociation of FBP17 from cortex, thus supporting the actin-mediated negative feedback proposed by our model.

Because F-BAR proteins could function in regulating actin dynamics in addition to their membrane-remodeling ability, the indispensable role of F-BAR in wave propagation did not necessarily prove a requirement for membrane shape changes. To further explore whether the wave propagation requires changes in membrane shape, we generated a series of domain-swapping mutant proteins that differed from FBP17 only in their curvature preferences and tested whether they could functionally replace FBP17 in the waves. Two types of mutants were designed based on biophysical properties of known curvature-generating proteins, which covered a range of curvature preferences with the absolute values from ~0.1 to ~0.01 nm$^{-1}$ (Fig. 3c). First, we replaced the F-BAR domain of FBP17 with other lipid-binding domains of different shapes: F-BAR domain of FCHo1; BAR domain of Endophilin2; and ENTH domain of Epsin1 (Fig. 3c, d). These domains represented three major classes of curvature-generating domains (F-BAR, BAR, and ENTH)[25,44,45]. Phospholipid-binding but curvature-insensitive PH domain of phospholipase C[46] or membrane-targeting sequence of tyrosine-protein kinase Lyn (Lyn10) were introduced as controls. Among these, the mutant with the constitutively membrane-targeting Lyn10 did not form waves in wild-type (WT) cells, consistent with a requirement of F-BAR dissociation from the membrane for waves (Fig. 3d, top). The rest of the domain-swapped mutants still localized to the waves in WT cells, indicating that these mutants were properly folded functional proteins. However, in DKD cells, only other F-BAR (F-BAR of FCHo1) could rescue wave formation (Fig. 3d, bottom). The fact that the BAR (from Endophilin2) or ENTH (from Epsin1) domain-swapped mutant could not rescue wave formation suggests that a specific range of curvature (similar to or lower than that of the F-BAR domains) is required for wave formation, as predicted (Fig. 3a).

For the second type of mutants, we introduced single point mutations to perturb the curvature preference of the F-BAR domain of FBP17 (Fig. 3c, e). Mutants K56E, K33E, and F276D had paralyzed tubule-generating ability in living cells[23,24]. Of particular interests are mutants K166A and K66E, which generated narrower (higher curvature) and wider tubules (lower curvature) compared with wild-type F-BAR domain, respectively, in in vitro liposome tubulation assays[24]. Both mutants K166A and K66E were localized in waves in WT cells (Fig. 3e, top), confirming that they were functional. However, only K66E point mutant of FBP17 (of shallower curvature preference) could rescue wave formation in DKD cells (Fig. 3e, bottom). Together with the domain-swapping data (Fig. 3d), these results indicate that the shape of the lipid-binding domain of FBP17 is important for wave formation, regardless of the origin of the curvature preferences being the intrinsic protein shapes or the conformation of protein oligomers. We conclude that membrane shape-mediated feedback is actively involved in the cortical wave propagation.

**Wave propagation requires optimal membrane deformability.** The requirement of membrane shape-mediated feedback confirmed the mechanochemical nature of the wave propagation. An independent test for this was on the mechanosensitivity of the waves. As membrane deformability critically determined rhythmic propagation of cortical protein recruitment in our model, the wave propagation was predicted to diminish with increasing or decreasing membrane tension (Fig. 4a). Hence, we investigated the effect of membrane mechanics on the waves. We conducted

osmolarity shock experiments by cyclically perfusing cells with hypotonic or hypertonic buffers; these treatments increased or decreased membrane tension, respectively, which subsequently affected membrane shape deformability. The kymographs show that osmolarity changes reversibly inhibited rhythmic wave propagation (Fig. 4b, Supplementary Movies 4 and 5). Osmotic treatment could lead to additional effects, including cell volume and cytosolic concentration changes (Supplementary Fig. 5). To circumvent these potential side effects, we used the surfactant deoxycholate (DC) to specifically reduce the plasma membrane tension without changing cell volume[47]. We observed that FBP17, actin, and membrane shape waves disappeared (Supplementary Fig. 6) in a DC dose-dependent manner (Fig. 4c). Such effects were reversible on the time scale of seconds upon DC washout, eliminating possibilities of permanent membrane compositional changes. Waves experiencing moderate DC or hyper-osmotic treatment displayed no significant changes in propagation speed, but prolonged oscillation periods (Fig. 4d), indicating that the membrane shape-mediated feedback quantitatively determines the cortical wave propagation.

**Pseudowave nature of curvature-sensitive protein wave.** So far, we demonstrated that membrane shape change and protein curvature sensitivity were integral to our mechanochemical wave. While the membrane undulation wave is mechanical, the cortical protein traveling waves are chemical waves. The key remaining questions are as follows: why is the wave speed (~µm s$^{-1}$) 10–100 times faster than other cortical actin waves? Does this ultrafast wave speed depend on the membrane shape-mediated feedback? What is the nature of such protein waves? In conventional reaction-diffusion systems (e.g., BZ reactions[48]), real wave and "pseudowave" represent two extremes in the spectrum of chemical traveling waves[49]. Here real wave is driven by diffusion—a real propagation of chemicals in physical space[1]. In contrast, pseudowave is not a real material propagation in space; instead, it reflects the spatial gradient in the timing of local excitation that gives the propagating impression. The question is: in the context of chemical traveling waves, is our cortical protein wave propagation more akin to real or pseudowave?

To gain insight in cortical protein traveling waves, we carried out more vigorous mathematical analysis of our system. By ignoring protein lateral diffusion along the cortex, we simplified the model to make it possible to obtain analytical solution of traveling wave. A nontrivial steady-state traveling wave solution emerged with a nonzero membrane curvature at the wavefront. This analytic solution recapitulates the salient features of traveling wave predicted by the full model, including modulation of the wave speed by protein curvature sensitivity and membrane mechanics (Supplementary Note 2).

Interestingly, without protein lateral diffusion, this analytical solution yielded similar wave speed as those from the full model (Fig. 5a). This finding suggests that diffusion need not be important in wave propagation, indicating that our protein traveling wave may not reflect real material movement in the direction of wave propagation. To further pinpoint the nature of our protein traveling wave, we borrowed the criteria that distinguishes pseudowave from real wave[49]: a wave is a pseudowave if the concentration changes—resulting from temporal chemical reaction at each point in space—are much larger in magnitude than those resulting from diffusion. Otherwise, it is a real wave with the propagation speed that scales as (protein diffusion constant)$^{1/2}$ [50]. In this regard, diffusion in our nominal case contributed little but negatively to the wave propagation over

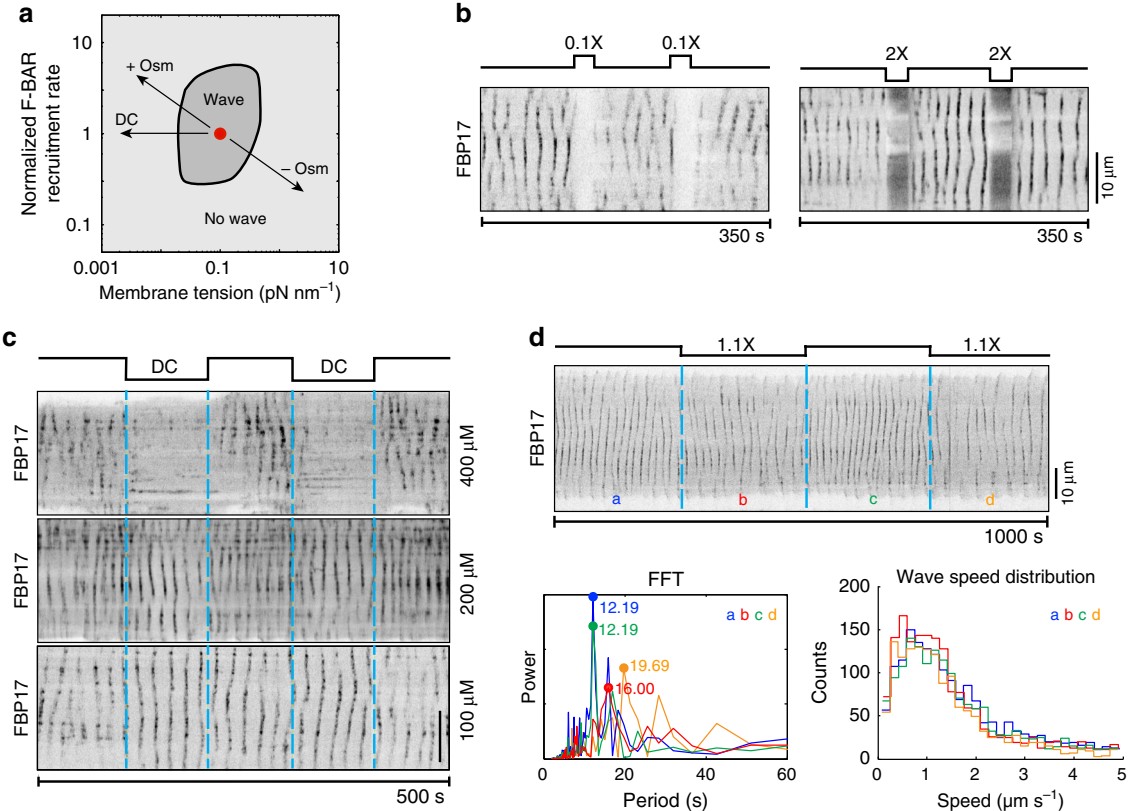

**Fig. 4** Membrane mechanics is essential for wave propagation. **a** Model phase diagram showing the dependence of traveling waves on the membrane tension and the F-BAR cortical recruitment rate. The red dot represents the model parameter set that generated the nominal model results in Fig. 2. **b** Osmotic shock reversibly inhibits traveling waves as shown by the FBP17-GFP kymographs. Cells were cyclically perfused with isotonic (300 mOsm) buffer for 100 s and hypotonic (30 mOsm) or hypertonic (600 mOsm) buffer for 25 s (Supplementary Movies 4 and 5). **c** Kymographs of FBP17-GFP show that surfactant deoxycholate (DC) reversibly inhibited traveling waves in a dose-dependent manner. Cells were cyclically perfused with isotonic buffer without DC for 100 s and isotonic buffer with DC at different concentrations (400, 200, and 100 μM, respectively) for 100 s. **d** Mild osmotic shock changes oscillation period but not wave speed. Top: kymograph of FBP17-GFP waves in a cell that was cyclically perfused with mild hyper-osmotic buffer for 250 s each duration. In duration a and c, the cell was perfused with 280 mOsm buffer, while in duration b and d, cells were perfused with 310 mOsm buffer. Bottom left: fast Fourier transform (FFT) shows the dominant periods in each duration. Bottom right: wave speed distribution in each duration. Colors of plots indicate durations as in top. For **b**–**d** all the schematics above kymographs illustrate putative changes in cell membrane tension. Scale bar: 10 μm

time (Fig. 5b). Importantly, the wave speed was sub-diffusive ~ (protein diffusion constant)$^\alpha$, where $0 \le \alpha < 1/2$ (Fig. 5c): namely, increasing the protein lateral diffusion constant had only a minor effect on the wave speed until it disrupted the wave (Fig. 5c). Taken together, we suggest that our cortical protein traveling wave is more in line with pseudowave. Importantly, this is the only type of cortical protein traveling wave in our system—a unique feature distinct from conventional reaction-diffusion systems—that typically host both real and pseudowaves.

We then asked: what underlies this unique feature? While all traveling waves must have the promotion zone ahead of the wavefront to confer propagation (Fig. 5d, e), a key difference in our case lies in a peculiar nature of curvature sensing (Fig. 5e). In conventional reaction-diffusion systems, as lateral diffusion advances the wavefront, autocatalytic reactions of the local chemicals always promote excitation in the direction of wave propagation, because the chemical concentrations are always positive (Fig. 5d). In contrast, the stimulatory elements in our system include not only conventional chemical autocatalytic reactions but also curvature sensing effects. The membrane curvature changes sign from negative to positive as one moves away from the wavefront in the direction of wave propagation (Fig. 5e). Because F-BAR only accumulates in the negative-curvature region, it is not recruited to the positive-curvature region due to the geometric mismatch. Consequently, the positive

membrane curvature itself defines an inhibitory zone ahead of the wavefront: once F-BAR arrives via lateral diffusion, it is "prompted" to release into cytoplasm (Fig. 5e). As such, the curvature sensing constrains the effect of protein diffusional drift on wave propagation (Fig. 5e), a key feature that is absent in conventional reaction-diffusion systems (Fig. 5d).

We therefore suggest that curvature sensing-mediated traveling waves of cortical proteins mainly reflect the local protein recruitment from cytoplasm, rather than the protein lateral diffusion (Fig. 5c). What really propagates here is the membrane undulation that induces local protein recruitment from cytoplasm, potentiating wave propagation like a ripple moving in a pond. For the corresponding cortical protein traveling wave, it is the spatial gradient in the timing of local excitations that gives the propagating impression. Hence, the smaller this spatial gradient, the smaller the relative timing in excitations and, hence, the faster the protein wave appears to propagate.

To test this prediction, we used TIRFM experiments to track wavefront propagation with high spatial-temporal accuracy. For a given wave, wave speed is heterogeneous. In most cases, wave is at a higher speed during the "collapsing" at the epicenter; and the speed decreases as wave spreads out (Supplementary Fig. 7). Instantaneous wave speeds can be as fast as 5 μm s$^{-1}$ (Fig. 5f, Supplementary Fig. 7). Even if it slows down during propagation, the wave speed can be ~1 μm s$^{-1}$ (e.g., Supplementary Fig. 7).

Importantly, our experiments show that the spatial gradient of cortical protein density at the wavefront inversely correlates with the wave speed (circles, Fig. 5f). Because the spatial gradient of cortical proteins reflects the relative timing of the local excitation (Fig. 1e), this inverse relationship is consistent with our prediction (solid line, Fig. 5f). More critically, a zoom-in kymograph shows that the individual FBP17 punctum undergoes cycles of assembly and disassembly during wave propagation, but do not notably move in space (Fig. 5g), as predicted by our model (Fig. 5b, c). We conclude that our cortical protein traveling waves mainly reflect the protein recruitment from cytoplasm, rather than, the lateral diffusional drift.

## Discussion

In this work, we provide some mechanistic insights into cortical traveling waves. Our work highlights the importance of membrane shape change in establishing a mechanical cue for cortical protein recruitment, which reciprocally governs cortical dynamics that culminate in traveling waves. Critically, we show that in a curvature sensing-driven rhythmic propagation, the cortical protein traveling wave results mainly from the local protein recruitment from the cytoplasm, rather from lateral diffusion.

Our cortical protein traveling waves have several unusual features. First, while akin to pseudowaves, they display some characters of real waves[1,51,52]: the cortical protein traveling waves

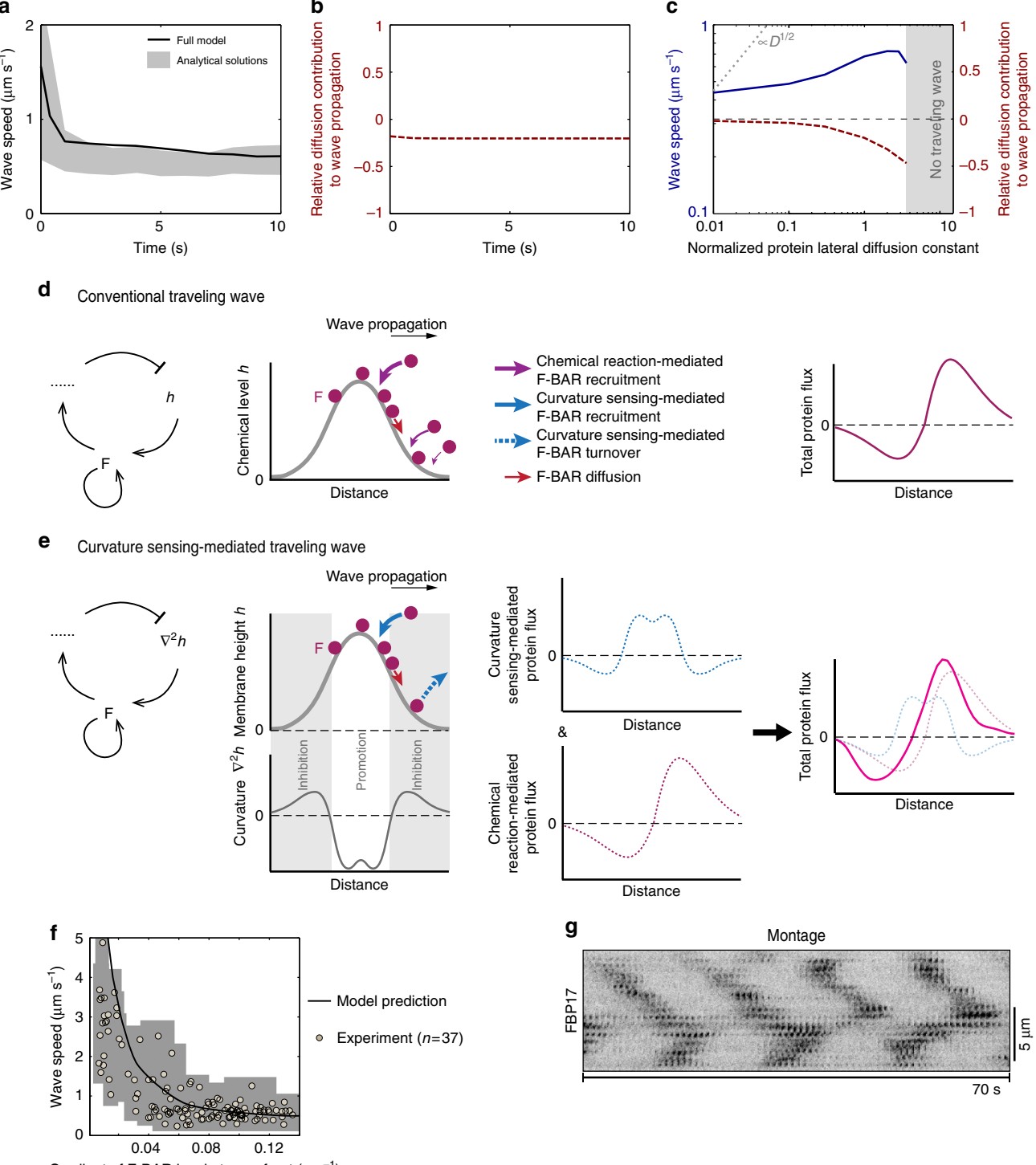

annihilate each other if they meet in a head-on collision, and deflect/split if they meet otherwise (Supplementary Fig. 10). This is because the protein traveling waves emerge from the membrane shape-mediated feedback, and are constrained by the accompanying membrane undulation that itself is a real wave. Second, while the speed of our mechanochemical wave is ultrafast, it is still modulated by the protein lateral diffusion along cortex. This is different from another interesting example of a mechanochemical wave[53], whose predicted wave speed is independent of protein lateral diffusion, as this chemical wave is entirely driven by and, hence, effectively reads out, membrane mechanics[53]. In contrast, in our model wave speed is controlled by the feedback between membrane curvature and cortical protein dynamics, and is not completely independent of protein lateral diffusion (Fig. 5c). Figure 5d, e shows that curvature sensing always constrains the effect of F-BAR lateral diffusion on wave propagation, and wave speed is always sub-diffusive, which has not been reported previously[53]. Third, in our model the sub-diffusive dependence of wave speed on protein lateral diffusion constant is a consequence of curvature sensing, not just because of the mechanochemical feedback. To demonstrate this, we altered the model in several ways, maintaining mechanochemical feedback but without curvature sensing, and found that the corresponding wave speed was not always sub-diffusive (Supplementary Fig. 11a, b, Supplementary Note 3). In contrast, the wave speed in the other model schemes—that preserve curvature sensing in the feedback—is always sub-diffusive (Fig. 5, Supplementary Fig. 11c–f, Supplementary Note 3).

Ultrafast wave speed (~µm s$^{-1}$) outruns the protein lateral diffusion and could enable cells to rapidly scan its surface for mechanical and chemical cues. Interference between traveling waves could make it possible for cells to coordinate and integrate cortical signals. While our model focuses on a specific system, it provides a general framework for curvature sensing-mediated traveling wave dynamics. The unusual features remain robust to variations in the detailed model scheme. They only require autocatalytic cortical recruitment of a curvature-sensitive protein coupled with a negative feedback (possibly involving actin dynamics) (Supplementary Fig. 11, Supplementary Note 3). Given that there are many other curvature-sensitive proteins involved in diverse cellular processes[5,18,54–56], the biological implications of our model can be broad. We note that curvature sensitive I-BAR domain proteins (e.g., ABBA[57]) interact with actin, and shape the micro ruffles in glial cells[56]. While ABBA prefers membrane protrusion, rather than inward bending by F-BAR, it could play a similar role as F-BAR, and mediate traveling wave-like behavior of the dorsal ruffles.

Our model recapitulates the basic features of our traveling wave but is inevitably incomplete. First, we focused on the cortex-centric mechanism of rhythmic propagation. An open question is: can the cortex-bound protein wave be part of a real wave in the cytoplasm? Even if the wave exists in the cytoplasm, we reason that the membrane shape change and the curvature sensitivity of F-BAR are still essential for this rhythmic propagation (Figs. 2–4). Future work will investigate the possibility of traveling waves with a cytosolic origin. Second, the model treated the ventral membrane at the cell edge as a clamped boundary. While external stimulation (i.e., activation of Cdc42) was only applied at the beginning of the simulation (Fig. 1b), it elicited long-lasting oscillatory dynamics: after the first wave propagates outward, new waves initiate at the original epicenter. This is because the membrane at the epicenter does not entirely relax back to the baseline after the protein wave has passed. It is this residual membrane shape deformation that serves as a cue to recruit F-BAR, which in turn initiates the next round of oscillation (Supplementary Fig. 2a, b). On the other hand, because the clamped boundary dampens the membrane shape changes, its effect propagates from the edge to the epicenter, where it flattens the residual membrane shape deformation over time. Without external activation signals, this flattening of the membrane eventually prevents a new round of F-BAR cortical recruitment and, hence, dampens the oscillation at long times. This is consistent with our observations that the waves are only on the cell ventral side and not strictly sustained oscillations (as indicated by Fig. 2, and see more details in Supplementary Figs. 12 and 13). Nevertheless, our key conclusion remains robust regardless whether the dynamics is a sustained or dampened oscillation. In reality, the cell edge may evolve over time; and membrane-substrate adhesion is probably not spatially uniform as assumed. Interestingly, a localized adhesion that clamped the membrane diverted the wave propagation (Supplementary Fig. 14), lending further support to the mechanochemical nature of our wave. In the future, we would like to systematically investigate how dynamics of cell edge movement and cell-substrate adhesions impact our traveling wave. Third, our model proposed that the membrane stiffening by actin accumulation prevented membrane deformation and hence turned off F-BAR recruitment. This is consistent with a negative role of actin in cortical wave observed previously[3], and supported by our observations that the actin accumulation negatively correlates with the membrane height change (Supplementary Fig. 15). Also, latrunculin A treatment has effects on traveling waves similar to hyper-osmolarity (Fig. 4d, Supplementary Fig. 16). However, this proposal does not exclude additional effects of actin. For instance, actin

**Fig. 5** Curvature sensing-mediated cortical protein waves always reflect protein recruitment from cytoplasm onto membrane. **a** Time curve of wave speed. The analytic result was obtained by computing the analytical solution (Eq. (2.24)), in which we took the membrane shape at the wavefront from the full model simulation as the input, and fixed all the other parameters in the formula in accordance to the corresponding full model case. We note that there is one parameter in the analytic formula that cannot be analytically derived from the reaction rates in the full model. Instead, we can numerically determine the range of its value (Supplementary Note 2), which reflects on the gray zone in the analytic results. **b** Time curve of diffusional contribution to wave propagation. **c** Diffusion-dependence of wave propagation. **b**, **c** Relative diffusion contribution to wave propagation is defined as the ratio between the diffusion term and the reaction term in the dynamic equation of F-BAR (Eq. (1.3) of the full model, Supplementary Note 1). And we chose the location ahead of the wavefront, where the absolute value of membrane curvature is the highest. Variation in this location does not change the qualitative conclusions from these plots. **d**, **e** Schematics of curvature sensing-mediated inhibitory effects on diffusional contribution to wave propagation. **d** Traveling wave from conventional reaction-diffusion system. **e** Curvature sensing-driven traveling wave. In **d** the "$h$" represents a chemical concentration in a generic sense, whereas the "$h$" in **e** is the membrane height and $\nabla^2 h$ is the membrane curvature. The round disk represents a cortical protein "F". We emphasize that the comparison here is only within the regime of traveling waves, not stationary wave phenomena that could emerge with characteristic spatial periodicities from conventional reaction-diffusion systems. **f** Inverse relationship between wave speed and the spatial gradient of cortical protein density at wavefront. The experimental data were collected from 37 waves in 8 cells. The F-BAR intensity for each wavefront was normalized to the maximum value within the individual cell. The gray shade represents the tracking measurement uncertainty (Supplementary Figs. 8 and 9). **g** Representative montage of FBP17 punctum during the apparent wave propagation show that each punctum assembles and disassembles at a fixed location

polymerization may push the membrane toward the substrate, increasing the membrane-substrate adhesion. We showed that our key conclusion was robust to this model variation (Supplementary Fig. 11c, Supplementary Note 3). Alternatively, actin retrograde flow could remove F-BAR from the membrane. Dissecting more detailed molecular events of actin machinery in our traveling waves will be part of our future work.

In sum, our work identifies membrane shape change as an important factor in determining the dynamics of traveling waves propagating along the cell cortex. This finding warrants closer scrutiny of the role of curvature sensing in other rhythmic cortical phenomena[2,3,8–16], possibly including actin-based cell polarity establishment[58–61] and cytokinesis[62].

## Methods

**Cell culture and transfection.** RBL-2H3 cells (tumor mast cells) were maintained in monolayer in minimum essential medium (Life Technologies, Carlsbad, CA) containing 20% fetal bovine serum (Sigma-Aldrich, St. Louis, MO) and 50 µg ml⁻¹ gentamicin (Life Technologies). Cells were harvested with TrypLE Express (Life Technologies) 2–5 days after passage. For transient transfections, electroporation with Neon transfection system (Life Technologies) was used. After transfection, cells were plated at subconfluent densities in 35-mm glass bottom dishes (MatTek, Ashland, MA) or on round coverslips in a 12-well plate overnight. Before imaging, cells were washed twice with Tyrodes buffer (135 mM NaCl, 5.0 mM KCl, 1.8 mM CaCl₂, 1.0 mM MgCl₂, 5.6 mM glucose, and 20 mM Hepes (pH 7.4)). For double-knockdown experiments, cells transfected with four HuSH-29 shRNAs (Origene, Rockville, MD) at 0.05 µg µl⁻¹ for each shRNA targeting rat CIP4 (TATGC-GAAGCAACTCAGGAGTCTGGTGAA, GCCGCAGAGTCCGTGGATGCTAA-GAACGA) and FBP17 (ATGGACGGCCGACATCAATGTGACCAAGGC, TGAAACGCACGGTGTCAGACAACAGCCTT) were incubated for 48 h before imaging. In all experiments, cells were sensitized with mouse monoclonal anti-2,4-dinitrophenyl IgE (Sigma-Aldrich) at 0.5 µg ml⁻¹ overnight and stimulated with 80 ng ml⁻¹ multivalent antigen, 2,4-dinitrophenylated bovine serum albumin (Life Technologies). For experiments using inhibitors, latrunculin A (Sigma-Aldrich) was diluted from the stock and added to the cells at a final concentration of 2 µM.

**Molecular cloning and plasmids.** Domain-swapping mutants of FBP17 were generated by replacing F-BAR domain amino acids (a.a.) 8–288 of human FBP17-EGFP with F-BAR domain a.a. 1–275 of mouse FCHo1, BAR domain a.a. 1–241 of rat Endophilin2, ENTH domain a.a. 1–144 of rat Epsin1, PH domain a.a. 1–170 of human PLCΔ1, membrane-targeting sequence a.a. 1–10 of human tyrosine-protein kinase Lyn, respectively. DNA sequences corresponding to a.a. 1–7, and a.a. 289–295 of FBP17 were kept as homologous regions for overlapping PCR. Truncation mutants and point mutations on F-BAR of FBP17 were generated using Phusion Site-Directed Mutagenesis Kit (Thermo Scientific, Waltham, MA). ΔF-BAR, ΔHR1, and ΔSH3 mutants were generated from mCherry-FBP17 by deleting a.a. 7–257, a.a. 406–483, and a.a. 554–608, respectively. Constructs for the following proteins were kind gifts: Lifeact-mRuby from Dr. Roland Wedlich-Soldner (Max Planck Institute of Biochemistry, Martinsried, Germany); FBP17-EGFP, mCherry-FBP17, CBD-GFP (a Cdc42 activity probe), and mCherry-actin were from Dr. Pietro De Camilli (Yale University School of Medicine, New Haven, CT) as previously described[4]. All plasmids were sequenced to vindicate their integrity.

**Total internal reflection fluorescence microscopy.** A Nikon Ti-E inverted microscope equipped with perfect focus system, iLAS2 motorized TIRF illuminator (Roper Scientific, Evry Cedex, France) and an Evolve 512 EMCCD camera (Photometrics, Tucson, AZ) was used. All images were acquired through a TIRF-CFI objective (Apochromat TIRF 100XH numerical aperture (NA) 1.49, Nikon). Samples were excited by 491 nm (100 mW) or 561 nm (100 mW) laser, reflected from a quad-bandpass dichroic mirror (Di01-R405/488/561/635, Semrock, Rochester, NY). The emitted light was acquired after passing through an emission filter (FF01-525/45 for GFP or FF01-609/54 for RFP/mCherry, Semrock) located on a Ludl emission filter wheel. MetaMorph 7.8 software (Molecular Device, Sunnyvale, CA) was used for image acquisition. Samples were maintained at 37 °C throughout the experiments using an on-stage incubator system (Live Cell Instrument, Seoul, South Korea).

**SRIC microscopy.** The same setup as above was used except that a mercury lamp (X-Cite 200DC, 200 W, Excelitas, Waltham, MA) was used as the light source. The mercury lamp had a spectrum output at wavelengths between 340 and 800 nm. A bandpass filter (609/54; Semrock) and neutral density filters (ND4) were used. Near-parallel rays produced by closing the aperture diaphragm obliquely illuminated the sample through the TIRF-CFI objective (100X N.A. 1.49, Nikon).

Reflected light rays from two interfaces (between coverslip and solution and between solution and plasma membrane) interfered with each other, creating a bright or dark patch when they were constructive or destructive, respectively. In order to perform sequential TIRF/SRIC with fast frame rate, mechanical changes of the optical parts were reduced to a minimum level. A 30/70 beam splitter was used at the back port of the L-shape fluo-illuminator of the Nikon microscope to combine the white light source (mercury lamp used for SRIC) and the lasers (used for TIRF). The same quad-bandpass dichroic cube (Di01-R405/488/561/635, Semrock) used for TIRF was used for SRIC, which allows an estimated 2% of incident light (609/54) to be reflected during the SRIC acquisition. Similar results were obtained using a Nikon SRIC cube composed of excitation filter 535/50, dichroic mirror DM400, and neutral density filter ND16.

**Perfusion experiments.** Transfected cells were seeded on 20 mm round coverslips in a 12-well plate overnight. Prior to imaging, the coverslip was transferred to a custom perfusion chamber (Chamlide, LCI) placed on the heated stage of a Nikon Ti-E inverted microscope. A multi-valve perfusion control system (MPS-8, LCI) was used to switch rapidly between solutions flowing into the chamber and over the cells. The perfusion system was connected to a computer and controlled by MetaMorph software (Molecular Device) in synchronization with image acquisition. A typical flow rate was ~0.3 ml min⁻¹. For the inner channel dimension of 10 mm ($L$) × 2 mm ($W$) × 0.2 mm ($H$), the flow rate was calculated to be 12.5 mm s⁻¹ in the channel, which means the solution in the chamber was fully exchanged in <1 s. The following solutions were used: Tyrodes buffer; Tyrodes buffer supplemented with 400 µM DC; hyper-osmotic Tyrodes buffer with 2X solute concentration; and hypo-osmotic Tyrodes buffer diluted with distilled deionized water to 0.1X/0.5X concentration. pH of all buffers was adjusted to 7.4. Osmolarity was confirmed using an osmometer (Osmomat 030, GONOTEC GmbH, Berlin, Germany).

**Image analysis.** An ImageJ-based software Fiji[63] was used to generate movies, kymographs, montages, and Z-stack projections. MatLab (MathWorks, Natick, MA) was used for data quantification and plotting. We determined the instantaneous wave speed and the cortical protein gradient at wavefront by the following procedures (as illustrated in Supplementary Figs. 8 and 9). To avoid complications arising from the interactions between multiple waves, we only chose well-defined single waves to determine the wave speed and the cortical protein gradient at the wavefront. We set the boundaries of individual waves by determining where the image intensity fell to below a threshold slightly above the background intensity level (Supplementary Fig. 8b); wave speed determinations were not sensitive to the threshold level (Supplementary Fig. 9a). Next, the wave centroid $r_c(t_i)$ of the segmented region for each time frame $t_i$ was calculated (Supplementary Fig. 8b). From the trajectory of the wave centroids over time, we obtained the instantaneous wave propagation direction, and wave speed, as determined by $v = |r_c(t_{i+1}) - r_c(t_i)|/(t_{i+1} - t_i)$ (Supplementary Fig. 8c). The line intensity profile of FBP17 along the wave propagation direction (the arrow in Supplementary Fig. 8c) was used to obtain the FBP17 gradient at wavefront (Supplementary Fig. 8d). The FBP17 gradient was calculated by $g = (I_{rc} - I_{bk})/I_{max}/d_g$. Here $I_{rc}$ was the intensity at centroid, $I_{bk}$ was the background intensity, $I_{max}$ was the maximum intensity of this measured cell, and $d_g$ was the distance from wave centroid to the wave edge where the intensity decreased to the background level (Supplementary Fig. 8d).

The robustness of our wave-tracking results depended on the chosen segmentation threshold value. To ensure that the tracking can give consistent results, we varied segmentation threshold values to determine the optimal range. Our calculation showed that at each time point of a given wave, the instantaneous wave speed from our tracking analysis varied with the segmentation threshold (step 1 in Supplementary Fig. 9a). We chose the segmentation threshold to be in a range within which the speed variation was minimal (step 2 in Supplementary Fig. 9a), which made our speed measurements converge. By overlapping the individual range from every wave at every time point of the same cell, the overall optimal threshold range for this cell was determined (steps 3 and 4 in Supplementary Fig. 9a). This optimal range could vary in different cells in part due to the different background intensities. Throughout the tracking, we thus chose a fixed segmentation threshold value within the optimal range that is independently determined for individual cells.

Further, to calculate the FBP17 gradient, we obtained its intensity profile along the wave propagation direction by averaging its intensity value around centroid within the box (step 1 in Supplementary Fig. 9b). Consequently, the measured FBP17 gradient was in subject to the box size (step 2 in Supplementary Fig. 9b). For each time point of an individual wave, the variation in the measured FBP17 gradient using different box sizes was then determined (step 3 in Supplementary Fig. 9b). By calculating all the gradient variations from different waves at different time points, we obtained the correlation between the uncertainty and the average of our measured gradient (steps 4 and 5 in Supplementary Fig. 9b). Importantly, the smaller the average gradient was the smaller was the uncertainty. The uncertainty in the FBP17 gradient measurement was plotted as the gray shadow in Fig. 5f.

**Theoretical model analysis**. Detailed theoretical model formulation and analysis are provided in Supplementary Notes 1–3, and Supplementary Tables 1–3.

**Data availability**. Data supporting the findings of this study are available within the article (and its Supplementary Information file) and from the corresponding authors on reasonable request.

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

## Acknowledgements

We would like to thank Michael Sheetz for suggesting the deoxycholate experiment, Yang Yang and Ding Xiong for sharing experimental results, and Emma Feng and Larry Cheung for technical assistance. We also thank Jonathan Silver for critical reading and editing. This work was supported by the intramural research program of NHLBI at NIH (J.L.) and the National Research Foundation (NRF) Singapore under its NRF Fellowship Programme (M.W., NRF Award No: NRF-NRFF2011-09).

## Author contributions

Z.W., M.S., C.T., M.W. and J.L.: conception and design of the project, acquisition of data, analysis and interpretation of data, and drafting or revising the article. Z.W. and M.S. have contributed equally to the work.

## Additional information

**Competing interests:** The authors declare no competing financial interests.

