## [Peer Review File · Nature Communications]

Reviewers' comments:

Reviewer #1(Remarks to the Author):

This paper by Wu et al. describes an interesting protein-affinity wave phenomenon which travels across the plasma membrane in cells. By using TIRF and reflectance microscopy the authors are able to detect nanoscale height changes in conjunction with protein intensity levels at the cell surface which is in contact with the substrate. The concept is that Cdc42, actin and FBAR are sufficient in generating a traveling phase wave across the cell surface. Importantly, the curvature sensing ability of the FBAR is found to be critical generating the wave.

This topic is highly under-represented in literature and thus this paper provides a much needed study of the protein dynamics which is responsible in forming the cell surface and in transmitting signals across the cell. The fact that the authors can experimentally reproduce their model proves that this wave property of protein binding to the surface (and corresponding deformation of the surface) is potentially an important property of the cell in e.g. mechano-signaling. I feel that the authors should try to motivate their study better in the introduction or discussion since this is a poorly studied area of great importance.

I would recommend publication of this work if another reviewer can verify the quality of the simulations which is not my area of expertise.

Comments:

Too little is written about the method in the paper. E.g. how is the experimental pulse generated? Clearly, explain the nature of this pulse. This should be explained in the text preceding figure 1. I had to read the whole paper including the SI before I properly understood the concept of the paper.

At line 99-101 it is postulated that curvature sculpting triggers further recruitment (=sensing). It would be appropriate to discuss the density dependence of curvature sensing in this context (as found in many bar domain papers (eg. Ref 22)). Deformation of membranes by bar domains occurs at high surface densities and the sensing is found to be very weak at such densities. Contrary, at low density, the sensing is strong, but the membrane sculpting ability of the protein is low at low density. This is a bit contradictory to what is stated above since sculpting would require high densities, but high densities would lead to low sensing.

A better explanation of the model would be beneficial- e.g. how is curvature sensing implemented in the model? A lot is written in supplementary, but reading the paper should give the reader an idea about the strategy.

What is the motivation for studying FBAR? A little discussion concerning different membrane proteins like IBARs (in particular ABBA) should be included. ABBA has also been found to be curvature sensitive, does shape the cell periphery (e.g. micro ruffles in glial cells) and does also interact with actin. This discussion could be included in introduction, discussion or at line 253-256 and would contribute to underscore the biological importance of this study.

On lines 158 and 159 Fig 1d is mentioned twice, is this correct?

On line 328 and 330: (protein diffusion) should be specified- I assume it is 'mean square displacement'.

On line 416: can a phase wave not carry information?

Figure 1e: how are the experimental f -bar levels measured. The ROIs (blue and green in (e)) look a little randomly inserted

Figure 2d: I cannot see the point of this merge-image. An overlay image should clearly reveal the colocalization or lack of same. All three images could have color which would help.

Figure 3d (caption) wave = wave formation

On line 198 : F-BAR ...what? ...density?

Reviewer #2 (Remarks to the Author):

The paper by Wu et al provides interesting advance into understanding of the earlier reported actin waves, which are associated with FBAR domain proteins. In the present manuscript the authors convincingly show the chemo-mechanical nature of these waves. They combine experiments and modeling to explain why the waves in this system appear to be faster than in other described systems. The paper is certainly interesting and provides a new advance in the field. However, there are serious flaws in the organization and presentation of the work, as well as in the interpretation of the results, that need to be addressed before the paper can be published. The authors emphasize that their model plays a central role, while experiments are mostly there to test the model. Therefore, the quality of modeling and its rigor play the decisive role in assessing the value of this paper. Unfortunately, in the present version many aspects of modeling are left to the readers to figure out by themselves:

1. cursory look at the model shows that the biochemical part of it is entirely heuristic. The authors postulate a positive feedback loop between Cdc42, FBAR domain proteins and N-WASP. This module drives accumulation of actin that provides negative feedback by stiffening the membrane. Given that the relationships between all the above molecules in the model are entirely heuristic and do not correspond to any specific molecular mechanism, it is puzzling that the authors need five (!) biochemical variables. All they need as one autocatalytic variable (the C-W-F module) that induces curvature and benefits from it and an inhibitory variable (A) that is induced by the activator. Normally, multiple variables arise because the authors attempt to incorporate all experimentally determined molecular interactions. The authors need to justify this entirely unwarranted proliferation of variables in the model that has no molecular mechanisms. Present set of variables is redundant from the mathematical point of view and provides false expectations among biologists that the model addresses specific molecular mechanisms that at the moment are not known.

2. More specifically, all biochemical equations in the model contain multiplicative term $(1-X)$, where $X=(C,W,F,R,A)$. On the surface, this term insures that none of the variables exceeds 1. However, it is possible that these terms also introduce additional nonlinearity into the model. The authors must justify these terms and provide mathematical evidence that their introduction is not absolutely necessary for the model to exhibit its present behavior.

3. Further to the biochemical mechanisms. The authors use fairly high orders of nonlinearity in the equations for R and A (2 and 4, respectively). Since the model is heuristic, the authors are free to choose. However, in equation for R they claim that "any cooperativity with Hill coefficient > 2 would preserve the model essence", but, nevertheless keep 4 (!), which is a lot higher degree of non-linearity. Why? The authors should justify why they use 4 when 2 is just fine or be honest about why 4, rather than 2, suits their purpose better and drop the above claim.

4. Fundamental issue with the interpretation of modeling results is a continuous swap between claiming oscillatory and excitable nature of the dynamics in the system. The authors talk about oscillations in some places and then about excitable dynamics in others. It is absolutely paramount for the paper to be published that the authors clearly identify the nature of their dynamics and show some proof. If it is oscillatory, they need to explain which part of the model drives periodic dynamics given that their waves die on the boundary due to the choice of the boundary conditions. If the system is excitable, they need to explain how exactly they generate periodic dynamics seen in figures 1d and 2c.

5. Presentation of modeling results – related to the above point 4. Figures 1 and 2 show extremely small domains of space on which the waves emerge from the center, presumably induced by the local perturbation shown in Fig. 1b, which argues against the oscillatory nature of the dynamics. The domain is so small that even one wavelength does not fit into it. The movie S2 does not help the situation, since the domain is the same. This makes the reader wonder if these waves are

indeed stationary phenomenon or only transients to some other type of dynamics. The caption in Fig. 2 says "for the purpose of demonstration (?), the snapshots... only show a part of simulated membrane patch". This is clearly not true for the Fig. 2a and Movie S2 because we see that the membrane is clamped at the boundary of the shown domain. This is surprising and makes the reader wonder if the authors hide some dynamics which they don't want the reader to see. Furthermore, Fig. 2c clearly suggests that the dynamics is not periodic, the profile of waves continuously changes and the last wave-length is clearly distinct from the middle. Is this because the waves are not stationary?

6. Related to the issues with presenting only snippets of dynamics. Figure S11 a(ii) shows a breakdown of a wave front. How was this result achieved? If this was done by introducing an inhomogeneity (obstacle), this should be clearly spelled out. If this is, in fact, instability of the front, this requires further explanation. Broken fronts (the amplitude is visibly different on the figure) are different dynamic patterns with potentially distinct dependence on diffusion coefficient, etc. If the front is intrinsically unstable, the simulations should be run on a large domain and the dynamics compared with that in experiments.

7. Another fundamental issue in this paper is the claim that the waves are "phase waves". The whole last section is devoted to a rather inherently inconsistent argument that the waves must be "phase waves". For example, the authors first state that the previously studied phase waves display interference, that is they penetrate through each other, but then they show that their waves do not have this property, they mutually annihilate (Fig. S11b). Strictly, phase waves are defined for oscillatory systems only and not for excitable systems. So, first the authors have to prove the oscillatory nature of their model. Furthermore, the term "phase waves" is not needed to explain why the speed of waves is not determined by the diffusion coefficient on the membrane. Indeed, the authors show that their waves are chemo-mechanical in nature, so the dominant term in their velocity is determined not by diffusion coefficient but by mechanical constants of the membrane. So the argument that the waves are not diffusion limited is perfectly consistent with chemo-mechanical nature of waves and does not require them to be phase waves. The authors should drop the confusing claim of "phase waves" as, in fact, they don't need this incorrectly used term to explain their results in both experiment and model. It is well known that chemo-mechanical waves are not restricted by diffusion coefficient (see e.g., PubmedID: 27074688).

8. In figure 5d and corresponding text, the authors claim that the difference between reaction-diffusion system and curvature sensing system is that the flux profile has a simple maximum in the first case and a complex profile with changing sign in the second case. This statement is entirely wrong, see, for example, Murray, *Mathematical Biology* 2nd ed, Ch 14.2. In both cases, to make a localized structure, the profile of the flux has to be positive (from cytoplasm on the membrane) in the center of the structure and negative (from membrane to the cytoplasm) on the sides. This factual error has to be corrected.

9. English language. Unfortunately, extremely poor English is not a minor problem of this manuscript but really makes it difficult to read and understand the paper. I cannot list all the instances because that would take many, many pages. Already in the abstract the authors mention "lateral diffusions" instead of "lateral diffusion coefficients". Many times plural form is used for other entities that cannot be plural, such as "feedbacks", "Discussions", etc. On page 6 the authors keep using non-existent expression "in pace", probably meaning to use "in phase". The language of the paper needs serious work before it is publishable.

Reviewer #3 (Remarks to the Author):

In the manuscript "Membrane shape-mediated wave propagation of cortical protein dynamics", the authors present their work on waves within the cortex of immune cells that travel much faster than typical cortical actin waves. While most studies in biophysics are either theoretical, computational, or experimental, the authors combine a model based on a set of differential equations with data from their own experiments. The results of the mechanochemical model reproduce the experimental data very well, and both experiment and theory complement each other. The authors conclude that "membrane shape changes and protein curvature sensitivity have underappreciated roles in setting high-speed cortical signal transduction rhythms". They propose that the unexpectedly high speed of the waves represents an efficient mechanism for cortical signal transmission.

In my opinion the theoretical model together with experimental tests appeals to readers of Nature Communications, even though the mathematical approach is rather simple and contains many independent parameters that are provided in a long table in the supporting information. As main outcome, the authors show diagrams that indicate in which cases waves are found depending on, for example, curvature-dependent F-BAR recruitment rates, characteristic F-BAR membrane curvature, and surface tension. They furthermore predict the wave speed numerically, even with the simplified analytical model, and propose that the membrane waves can transport information within cells faster than diffusion. This is to my knowledge a novel hypothesis that can strongly influence the field. However, I have a couple of issues mainly related with the modeling that I would like to ask the authors to address:

For the microscopy the membrane is close to a substrate, which is taken into account by a homogeneous adhesion energy to the substrate in Eq. (2.6). I suppose that this parameter is not well known and I wonder how much the results are sensitive to it. Furthermore, is it justified to assume a spatially homogeneous adhesion energy for the cell-substrate interaction?

In the simulation a free boundary condition is imposed. In the experiments, as for example seen in Fig. 1(e), the area where the cell is close to the substrate is finite. In which way do the authors expect the boundary condition to affect the wave formation?

The set of differential equations that the authors solve contains a term that couples the membrane shape to the concentration of the curved F-BAR proteins in Eq. (2.6), while a term that couples F-BAR motion within the lipid bilayer to the membrane curvature is missing in Eq. (2.3). Such a directed motion when the proteins follow the bilayer deformation can be expected to be much faster than diffusion of proteins in the bilayer. The authors have shown that diffusion can be neglected, but are there arguments that could indicate why the coupling of proteins to the membrane curvature occurs only via attachment of proteins from the cytosol (and not via curvature-driven motion of the proteins within the bilayer)?

I would have expected that in immune cells the actin dynamics would also couple directly to the membrane shape dynamics and not only via an effective surface tension. However, this is not taken into account in Eqs. (2.1) to (2.6). Can the authors comment on this?

Regarding the conclusions, a wave that travels faster than the speed of light cannot convey information. I am not sure whether this is what the authors write in the last paragraph of the main text? Also, I do not understand whether a chain of lamps flashing in quick succession should be an example for a wave that is faster than the speed of light? I recommend that the authors reconsider the wording in this paragraph.

Response to Reviewers' comments:

Reviewer #1 (Remarks to the Author):

This paper by Wu et al. describes an interesting protein-affinity wave phenomenon which travels across the plasma membrane in cells. By using TIRF and reflectance microscopy the authors are able to detect nanoscale height changes in conjunction with protein intensity levels at the cell surface which is in contact with the substrate. The concept is that Cdc42, actin and FBAR are sufficient in generating a traveling phase wave across the cell surface. Importantly, the curvature sensing ability of the FBAR is found to be critical generating the wave.

This topic is highly under-represented in literature and thus this paper provides a much needed study of the protein dynamics which is responsible in forming the cell surface and in transmitting signals across the cell. The fact that the authors can experimentally reproduce their model proves that this wave property of protein binding to the surface (and corresponding deformation of the surface) is potentially an important property of the cell in e.g. mechano-signaling. I feel that the authors should try to motivate their study better in the introduction or discussion since this is a poorly studied area of great importance.

I would recommend publication of this work if another reviewer can verify the quality of the simulations which is not my area of expertise.

Comments:

Too little is written about the method in the paper. E.g. how is the experimental pulse generated? Clearly, explain the nature of this pulse. This should be explained in the text preceding figure 1. I had to read the whole paper including the SI before I properly understood the concept of the paper.

Response 1.1: In our experiments, the waves emerge on the cell ventral surface after multivalent antigen cross-linking of Fc-receptors on the mast cell surface. We previously demonstrated that such antigen stimulation elicits transient local activation of Cdc42, which serves an upstream signal that facilitates the recruitment of FBP17 at the plasma membrane (Wu et al., PNAS, 2013). In the revised manuscript, we introduced the experimental pulse generation, setting the stage for the results in Fig. 1 (line 84-92 and 139-140).

At line 99-101 it is postulated that curvature sculpting triggers further recruitment (=sensing). It would be appropriate to discuss the density dependence of curvature sensing in this context (as found in many bar domain papers (eg. Ref 22)). Deformation of membranes by bar domains occurs at high surface densities and the sensing is found to be very weak at such densities. Contrary, at low density, the sensing is strong, but the membrane sculpting ability of the protein is low at low density. This is a bit contradictory to what is stated above since sculpting would require high densities, but high densities would lead to low sensing.

Response 1.2: We thank the reviewer for pointing out this importance issue. The membrane shape change in our system only involves shallow undulations, as evidenced by our SRIC experiment. This suggests low concentrations of F-BAR domain proteins on the cortex, which, otherwise, will highly deform the membrane into narrow tubules as many *in vitro* experiments demonstrate. Our model therefore concerns the low F-BAR concentration limit. In this limit, as the reviewer noted, the curvature-sensing effect is strong, whereas the curvature sculpting activity is weak. Accordingly, to capture

the weak curvature sculpting activity, our mathematical model only describes that the membrane-binding of F-BAR domain proteins moderately increases the local membrane height, rather than causing membrane tubulation. This way, F-BAR domain proteins only cause the membrane undulation, instead of deforming the membrane into tubules. We now realized that the phrase “curvature sculpting” perhaps specifically refers to membrane tubulation activities. To avoid this confusion, we replaced the phrase of “curvature sculpting” or alike by “membrane shape deforming”. In the revision, we further clarified the precise meaning of membrane shape deforming (line 98-105).

A better explanation of the model would be beneficial- e.g. how is curvature sensing implemented in the model? A lot is written in supplementary, but reading the paper should give the reader an idea about the strategy.

Response 1.3: In the new draft, we re-wrote the paragraph depicting the mathematical formulation of the model more explicitly (line 118-130). In particular, we modeled the dynamics of F-BAR to the cortex as a reaction-diffusion process, wherein the local mean curvature of the membrane increases the recruitment rate of F-BAR, *aka* curvature-sensing effects. Furthermore, the local membrane binding of F-BAR causes the local membrane to bend inward moderately, increasing the membrane height. This way, the curvature sensing and membrane shape deformation form a closed feedback loop.

What is the motivation for studying FBAR? A little discussion concerning different membrane proteins like IBARs (in particular ABBA) should be included. ABBA has also been found to be curvature sensitive, does shape the cell periphery (e.g. micro ruffles in glial cells) and does also interact with actin. This discussion could be included in introduction, discussion or at line 253-256 and would contribute to underscore the biological importance of this study.

Response 1.4: We thank the reviewer for providing the nice point for discussion. The implications of our work could shed light on other systems, such as ruffling in glial cells. In this case, I-BAR domain proteins – while prefers the membrane protrusion, rather than inward bending – could play a similar role as F-BAR domain proteins, and mediate wave-like behavior of the ruffles. Indeed, incorporating this elaboration in our new draft further underscores the biological importance of our study (line 431-436).

On lines 158 and 159 Fig 1d is mentioned twice, is this correct?

Response 1.5: We got rid of the first “fig. 1d” in the sentence.

On line 328 and 330: (protein diffusion) should be specified- I assume it is ‘mean square displacement’.

Response 1.6: We fixed this textual error. It should be “protein diffusion constant”.

On line 416: can a phase wave not carry information?

Response 1.7: We realized that such argument is not necessary and causes confusion. We therefore deleted the relevant discussion about phase wave in the revision.

Figure 1e: how are the experimental f-bar levels measured. The ROIs (blue and green in (e)) look a little randomly inserted.

Response 1.8: We subtracted the FBP17 fluorescence intensity by the background, and then normalized it by the maximum value. The two ROIs are chosen to show that oscillations at the different locations have stable phase shift. In fact, this feature persists for any two ROIs along the direction of wave propagations. We amended this elaboration in the revised main text (line 729-734).

Figure 2d: I cannot see the point of this merge-image. An overlay image should clearly reveal the colocalization or lack of same. All three images could have color which would help.

Response 1.9: We meant to show that membrane height changes couple with the FBP17 wave propagation. While the two wave patterns do not colocalize at the pixel level, they overlap and follow each other closely in time and space. We updated the images with colors and now the new Fig.2d shows the correlation more clearly.

Figure 3d (caption) wave = wave formation

Response 1.10: Yes, it should be “wave formation” (line 778-785).

On line 198 : F-BAR ...what? ...density?

Response 1.11: We fixed this as “local cortical F-BAR density” (line 199).

Reviewer #2 (Remarks to the Author):

The paper by Wu et al provides interesting advance into understanding of the earlier reported actin waves, which are associated with FBAR domain proteins. In the present manuscript the authors convincingly show the chemo-mechanical nature of these waves. They combine experiments and modeling to explain why the waves in this system appear to be faster than in other described systems. The paper is certainly interesting and provides a new advance in the field. However, there are serious flaws in the organization and presentation of the work, as well as in the interpretation of the results, that need to be addressed before the paper can be published.

The authors emphasize that their model plays a central role, while experiments are mostly there to test the

model. Therefore, the quality of modeling and its rigor play the decisive role in assessing the value of this paper. Unfortunately, in the present version many aspects of modeling are left to the readers to figure out by themselves:

1. cursory look at the model shows that the biochemical part of it is entirely heuristic. The authors postulate a positive feedback loop between Cdc42, F-BAR domain proteins and N-WASP. This module drives accumulation of actin that provides negative feedback by stiffening the membrane. Given that the relationships between all the above molecules in the model are entirely heuristic and do not correspond to any specific molecular mechanism, it is puzzling that the authors need five (!) biochemical variables. All they need as one autocatalytic variable (the C-W-F module) that induces curvature and benefits from it and an inhibitory variable (A) that is induced by the activator. Normally, multiple variables arise because the authors attempt to incorporate all experimentally determined molecular interactions. The authors need to justify this entirely unwarranted proliferation of variables in the model that has no molecular mechanisms. Present set of variables is redundant from the mathematical point of view and provides false expectations among biologists that the model addresses specific molecular mechanisms that at the moment are not known.

Response 2.1: Our traveling wave is the first reported case that couples actin wave with rhythmic propagations of curvature-sensitive proteins. We feel this is an observation that biologists are very interested in. Therefore, rather than constructing the simplest mathematical model, we aimed to distill the simplest possible mechanism of our traveling wave while preserving the essence of this novel coupling between actin machinery and membrane trafficking proteins. Otherwise, it would be difficult to faithfully compare our model results with our current experimental testing. We regret that we did not convey this point explicitly; in the revised manuscript, we further clarified this point.

Specifically, the mechanistic coupling between the actin machinery and the curvature-sensitive proteins (e.g., F-BAR) lies in the molecular interactions between Cdc42, N-WASP, and F-BAR domain proteins, which have been extensively documented *in vitro* and *in vivo*. Experiments show that F-BAR domain proteins not only directly bind to Cdc42 through its HR1 domain, but also bind to N-WASP. Such tripartite binding is essential for the axis of Cdc42–N-WASP–Arp2/3–actin polymerization (Ho et al., *Cell*, 2004; Takano et al, *EMBO J.*, 2008, cited as the references #22 and #33 in the paper). Moreover, our previous experiments show that Cdc42, N-WASP and F-BAR travel together in time and space during the wave propagation, whereas F-actin wavefront is ~5 seconds delayed (Wu et al., *PNAS*, 2013, cited as the reference #4). These evidences along with other experimental observations support a positive feedback loop between rapid cortical recruitments of Cdc42, F-BAR, and N-WASP.

We tried our best to incorporate these established molecular interactions into our model. In this regard, we respectfully disagree with the view that the biochemical part of our model is entirely heuristic. In fact, to build the model, we extensively analyzed literature data – on the specific physicochemical properties of, and interactions between, F-BAR proteins, actin machinery, and membrane – to obtain quantitative estimates for a range of relevant model parameters. Given the complexity of any biological systems, however, we are aware of the possibility that these molecular interactions – although essential and necessary – may not fully recapitulate the reality. We therefore devoted an entire

paragraph on the future improvement of our model in the discussion section (line 438-468).

On the other hand, the reviewer did make a valid point: The mechanism of a curvature sensing-mediated traveling wave could be further simplified from our current model. This touches upon the fundamental mathematical structure for the nature of curvature sensing traveling wave. We have already endeavored our efforts along this line: By combining Cdc42, N-WASP, and F-BAR as a single autocatalytic component, we successfully obtained our analytical solution of steady-state traveling wave in our system. In the revision, we further carried out detailed numerical simulations of this simplified model as the reviewer suggested and the alternative scenarios, some of which were suggested by the reviewer #3. Our results show that as long as curvature sensing is essential in the mechanochemical feedback, these model simplifications/alterations do not change the key conclusions. The key conclusions include 1) the coupling of membrane undulation with the protein traveling waves, and 2) the protein wave propagation always reflects local protein assembly from cytoplasm, rather than lateral diffusion along the cortex. In the revision, we elaborated this point in the discussion section (line 399-431), substantiated by additional results (the new supplementary Fig. 11).

While our current model starts from a specific case, equipped with these additional results, our work points to a more general framework for curvature sensing-mediated traveling waves. This mechanism could apply to other model systems: For instance, similar principle could play a role in dorsal ruffling in glial cells, where I-BAR domain proteins prefer membrane protrusion, instead of inward bending favored by F-BAR domain proteins. This points to a greater importance in biology. We further elaborated this point in the revision (line 432-436).

2. More specifically, all biochemical equations in the model contain multiplicative term $(1-X)$, where $X=(C,W,F,R,A)$. On the surface, this term insures that none of the variables exceeds 1. However, it is possible that these terms also introduce additional nonlinearity into the model. The authors must justify these terms and provide mathematical evidence that their introduction is not absolutely necessary for the model to exhibit its present behavior.

Response 2.2: The model variables, $X= (C, W, F, R, A)$, is normalized with 1 being their respective maximum densities. Therefore, the term $(1-X)$ ensures that each protein component does not accumulate beyond its own maximum density on the cortex, which makes physical sense. Nevertheless, when the $(1-X)$ term is replaced by 1, where $X=(C, W, F, R, A)$, the essence of our results does not change (please see the new supplementary Fig. 11e).

3. Further to the biochemical mechanisms. The authors use fairly high orders of nonlinearity in the equations for R and A (2 and 4, respectively). Since the model is heuristic, the authors are free to choose. However, in equation for R they claim that “any cooperativity with Hill coefficient > 2 would preserve the model essence”, but, nevertheless keep 4 (!), which is a lot higher degree of non-linearity. Why? The

authors should justify why they use 4 when 2 is just fine or be honest about why 4, rather than 2, suits their purpose better and drop the above claim.

Response 2.3: Our model with the Hill coefficient = 2 works just fine (please see the new supplementary Fig. 11f).

We chose Hill coefficient = 4 in our nominal case, because we'd expect even more nonlinearity in the Arp2/3-mediated actin polymerization in reality, as it is a highly complex and autocatalytic process (Pollard et al., *Annual Review of Biophysics and Biomolecular Structure*, 2000). For instance, branched actin network polymerization itself could provide more F-actin plus ends, which could locally accumulate more Arp2/3-complex. Alternatively, there are many other players in actin machinery could participate/feed back with Arp2/3-mediated actin polymerization. For example, WIP is another important player in the axis of N-WASP – Arp2/3 – branching actin polymerization: WIP is activated by N-WASP, and subsequently facilitates Arp2/3-mediated actin polymerization (e.g., Takano K et al., *EMBO J*, 2008). And another actin nucleation factor, formin, could also participate in Arp2/3-mediated actin polymerization (Block J, et al., *Current Biology*, 2012). These additional players will certainly increase the overall nonlinearity in the dynamics of actin polymerization; the Hill coefficient is therefore expected to be higher than 2 in reality.

The bottom line is: While the model does not mean to exhaust the molecular interactions in branching actin polymerization, our key conclusion does not hinge on this level of details. We further clarified this confusion in the revised supplement (SI, pg.6-7).

4. Fundamental issue with the interpretation of modeling results is a continuous swap between claiming oscillatory and excitable nature of the dynamics in the system. The authors talk about oscillations in some places and then about excitable dynamics in others. It is absolutely paramount for the paper to be published that the authors clearly identify the nature of their dynamics and show some proof. If it is oscillatory, they need to explain which part of the model drives periodic dynamics given that their waves die on the boundary due to the choice of the boundary conditions. If the system is excitable, they need to explain how exactly they generate periodic dynamics seen in figures 1d and 2c.

Response 2.4: We thank the reviewer for pointing out this confusion. We now clarified it. Briefly, although the dynamics in our current system is oscillatory, it gets dampened at long times by the boundary effect.

As evidenced in our previous experiments, the initial activation signal of the wave is local and transient. When this initial activation signal is sufficient, it drives oscillation. This is because the curvature-mediated positive feedback with F-BAR recruitment is intertwined by actin polymerization-mediated negative feedback with the curvature. As it takes a while (~ 5-7 seconds) to activate Arp2/3 due to the autocatalytic nature, it introduces a time delay in the actin-mediated negative feedback. It is this time delay in the negative feedback that causes a phase shift in actin dynamics with other components in the system and, hence, underlies the oscillation. As such oscillatory wave propagates and approaches the boundary of the system, it was disrupted by the

boundary condition, which clamps the membrane at the edge. While consistent with the experimental observation that the wave disappears at the edge of the cell ventral side, this boundary effect dampens the oscillatory dynamics in the longer timescale. This is because the activation signal is only transiently introduced at the epicenter at the beginning, whereas the effect of this clamped boundary persists and, hence, dampens the oscillations, disrupting the wave at long times. The smaller the system is, the faster the dampening in oscillation occurs. With a system size of 40 microns across, noticeable dampening in the oscillation at the epicenter starts to manifest in about 3 min in our current parameter set. This is evidenced in Fig. 2c, as the reviewer pointed out. Therefore, strictly speaking, the dynamics of individual wave in our system is not sustained but dampened oscillation. This notion is consistent with our experimental observation, in which single traveling wave – before interfered by other waves – dies out itself over time. However, we'd would like to point out that, regardless whether the dynamics is sustained or dampened oscillation, our key conclusions always hold, including 1) the coupling of membrane undulation with the protein traveling waves, and 2) the protein wave propagation always reflects local protein recruitment from cytoplasm, rather than lateral diffusions along cortex.

In the revision, we 1) removed the discussions/statements about excitable dynamics/medium etc to clarify the confusion; 2) elaborated the point about dampened oscillation on line 444-456; 3) amended the new supplementary Fig. 12 to demonstrate the dampening effects by the boundary; and 4) added the new supplementary Fig. 13 showing a representative experimental observation that the traveling wave dynamics is not a sustained oscillation.

5. Presentation of modeling results – related to the above point 4. Figures 1 and 2 show extremely small domains of space on which the waves emerge from the center, presumably induced by the local perturbation shown in Fig. 1b, which argues against the oscillatory nature of the dynamics. The domain is so small that even one wavelength does not fit into it. The movie S2 does not help the situation, since the domain is the same. This makes the reader wonder if these waves are indeed stationary phenomenon or only transients to some other type of dynamics. The caption in Fig. 2 says “for the purpose of demonstration (?), the snapshots... only show a part of simulated membrane patch”. This is clearly not true for the Fig. 2a and Movie S2 because we see that the membrane is clamped at the boundary of the shown domain. This is surprising and makes the reader wonder if the authors hide some dynamics which they don't want the reader to see. Furthermore, Fig. 2c clearly suggests that the dynamics is not periodic, the profile of waves continuously changes and the last wave-length is clearly distinct from the middle. Is this because the waves are not stationary?

Response 2.5: We updated the relevant figures/movie with larger ROIs that provide a better view of the dynamics. Our waves are traveling waves, not stationary phenomena. Furthermore, as discussed in the Response 2.4, the clamped membrane at the boundary dampens the oscillation and, hence, the amplitude of the wavefront declines in the long timescale. In the revision, we amended a new supplementary Fig. 2 for a global view of the same process.

6. Related to the issues with presenting only snippets of dynamics. Figure S11 a(ii) shows a breakdown of

a wave front. How was this result achieved? If this was done by introducing an inhomogeneity (obstacle), this should be clearly spelled out. If this is, in fact, instability of the front, this requires further explanation. Broken fronts (the amplitude is visibly different on the figure) are different dynamic patterns with potentially distinct dependence on diffusion coefficient, etc. If the front is intrinsically unstable, the simulations should be run on a large domain and the dynamics compared with that in experiments.

Response 2.6: For the Fig. S11 (now the supplementary Fig. 10 in the revised draft), the traveling wave is the same as the nominal case shown in Fig. 1E and, therefore, is stable. At the locations marked as red arrows in supplementary Fig. 10a(ii), we introduced a transient perturbation (time = 5 to 15 s) that directly quenched the local membrane shape deformation, which mimicked the effects of stochastic fluctuations. It is such a perturbation that breaks down the wave front, causing the splitting. We further elaborated the simulation details for the revised figure caption for the new supplementary Fig. 10.

7. Another fundamental issue in this paper is the claim that the waves are “phase waves”. The whole last section is devoted to a rather inherently inconsistent argument that the waves must be “phase waves”. For example, the authors first state that the previously studied phase waves display interference, that is they penetrate through each other, but then they show that their waves do not have this property, they mutually annihilate (Fig. S11b). Strictly, phase waves are defined for oscillatory systems only and not for excitable systems. So, first the authors have to prove the oscillatory nature of their model. Furthermore, the term “phase waves” is not needed to explain why the speed of waves is not determined by the diffusion coefficient on the membrane. Indeed, the authors show that their waves are chemo-mechanical in nature, so the dominant term in their velocity is determined not by diffusion coefficient but by mechanical constants of the membrane. So the argument that the waves are not diffusion limited is perfectly consistent with chemo-mechanical nature of waves and does not require them to be phase waves. The authors should drop the confusing claim of “phase waves” as, in fact, they don’t need this incorrectly used term to explain their results in both experiment and model. It is well known that chemo-mechanical waves are not restricted by diffusion coefficient (see e.g., PubmedID: 27074688).

Response 2.7: We agree with the reviewer that the phase wave argument is not necessary. In our new manuscript, we stopped claiming our wave as phase wave, as our system is not a sustained oscillatory system anyway. Instead, we emphasized that the cortical protein wave propagation mainly reflects the local protein recruitment from the cytoplasm, rather than the lateral diffusion.

As the reviewer pointed out, there is one example of mechanochemical wave in which the wave speed is predicted to be independent of the lateral diffusion constant of the chemicals. We now cited the work in the revised draft. Here, we’d like to note some interesting aspects of our traveling wave. While the speed of our mechanochemical wave is ultrafast, it is modulated by the protein lateral diffusion along cortex. This is different from the other interesting example of mechanochemical waves. In that case, the predicted wave speed is independent of the protein lateral diffusion, as the chemical wave is entirely driven by and, hence, is just a read-out of, membrane mechanics. In contrast, in our model wave speed is controlled by the feedback between membrane curvature and cortical protein dynamics. Fig. 5c shows that when the protein lateral diffusion constant is small, our wave speed is nearly independent of the diffusion,

similar to other mechanochemical waves. As the protein lateral diffusion constant increases, the wave propagation speeds up, but sub-diffusively. If the protein lateral diffusion constant is very large, the cortical protein profile and the membrane shape flatten, and the wave disappears. Fig. 5d shows that curvature sensing always constrains the effect of F-BAR lateral diffusion on wave propagation. These peculiar features are not reported in the previous work. Third, in our model the sub-diffusive dependence of wave speed on protein lateral diffusion constant is a consequence of curvature affecting reaction kinetics, not just because of the mechanochemical feedback. To demonstrate this, we altered the model, maintaining mechanochemical feedback but without curvature sensing, and found that the corresponding wave speed was not always sub-diffusive. In contrast, the wave speed in the other model schemes – that preserve curvature-sensing in the feedback – is always sub-diffusive.

In the updated manuscript, we elaborated this point in discussion section (line 400-420, the new supplementary Fig. 11).

8. In figure 5d and corresponding text, the authors claim that the difference between reaction-diffusion system and curvature sensing system is that the flux profile has a simple maximum in the first case and a complex profile with changing sign in the second case. This statement is entirely wrong, see, for example, Murray, *Mathematical Biology* 2nd ed, Ch 14.2. In both cases, to make a localized structure, the profile of the flux has to be positive (from cytoplasm on the membrane) in the center of the structure and negative (from membrane to the cytoplasm) on the sides. This factual error has to be corrected.

Response 2.8: We apologize for not making the point clear. For typical reaction-diffusion systems, it can display stationary waves – Turing patterns with characteristic domain sizes and hence stable spatial periodicities. Alternatively, they can exhibit traveling waves that do not necessarily have characteristic wavelength. What the reviewer referred here as “a localized structure” pertains to the former case. Since our waves are traveling waves and not stationary phenomena, we only meant to make the comparison within the regime of traveling waves in our previous draft.

The right panel in our original Fig. 5d only schematized the peculiar effect of curvature sensing on the F-BAR recruitment – defining an inhibitor zone ahead of the wavefront in the direction of wave propagation. It is this inhibitory zone that constrains the effect of diffusive fluxes on wave propagation. Here, we meant to emphasize the unique effect of curvature sensing, rather than providing the overall profile of F-BAR recruitment. We now realized this confusion and revised the Fig. 5d accordingly. Specifically, while influenced by curvature sensing, the F-BAR recruitment is also promoted by the pure chemical reactions, *i.e.*, the autocatalytic reactions between F-BAR, N-WASP, and Cdc42. These chemical reactions – like in any conventional chemical traveling waves – always define a promotion zone ahead of the wavefront to confer wave propagation. In our system, it is the overlapping of the effects from curvature sensing and pure chemical reactions that together defines the overall profile of F-BAR recruitment flux at the wavefront. As such, in our traveling wave the overall F-BAR recruitment is still promoted ahead of the wavefront. It is just that due to the constraining effect of curvature sensing, the wave speed is always sub-diffusive in our system. In contrast, conventional traveling

waves from reaction-diffusion systems do not have the constraining effect of curvature sensing; consequently, the protein lateral diffusion could manifest its full term in driving the wave propagation.

In the revision, we incorporated the essence of the above clarification on line 342-357, and updated the Fig. 5d along with the figure caption accordingly.

9. English language. Unfortunately, extremely poor English is not a minor problem of this manuscript but really makes it difficult to read and understand the paper. I cannot list all the instances because that would take many, many pages. Already in the abstract the authors mention “lateral diffusions” instead of “lateral diffusion coefficients”. Many times plural form is used for other entities that cannot be plural, such as “feedbacks”, “Discussions”, etc. On page 6 the authors keep using non-existent expression “in pace”, probably meaning to use “in phase”. The language of the paper needs serious work before it is publishable.

Response 2.9: We endeavored to improve the language and fixed the textual issues throughout the paper.

Reviewer #3 (Remarks to the Author):

In the manuscript "Membrane shape-mediated wave propagation of cortical protein dynamics", the authors present their work on waves within the cortex of immune cells that travel much faster than typical cortical actin waves. While most studies in biophysics are either theoretical, computational, or experimental, the authors combine a model based on a set of differential equations with data from their own experiments. The results of the mechanochemical model reproduce the experimental data very well, and both experiment and theory complement each other. The authors conclude that "membrane shape changes and protein curvature sensitivity have underappreciated roles in setting high-speed cortical signal transduction rhythms". They propose that the unexpectedly high speed of the waves represents an efficient mechanism for cortical signal transmission.

In my opinion the theoretical model together with experimental tests appeals to readers of Nature Communications, even though the mathematical approach is rather simple and contains many independent parameters that are provided in a long table in the supporting information. As main outcome, the authors show diagrams that indicate in which cases waves are found depending on, for example, curvature-dependent F-BAR recruitment rates, characteristic F-BAR membrane curvature, and surface tension. They furthermore predict the wave speed numerically, even with the simplified analytical model, and propose that the membrane waves can transport information within cells faster than diffusion. This is to my knowledge a novel hypothesis that can strongly influence the field. However, I have a couple of issues mainly related with the modeling that I would like to ask the authors to address:

For the microscopy the membrane is close to a substrate, which is taken into account by a homogeneous adhesion energy to the substrate in Eq. (2.6). I suppose that this parameter is not well known and I wonder how much the results are sensitive to it. Furthermore, is it justified to assume a spatially homogeneous adhesion energy for the cell-substrate interaction?

Response 3.1: We have done the phase diagram study on how the wave formation depends on the membrane-adhesion energy (supplementary Fig. 1c), which suggests

that qualitative features of our model results are preserved within a broad range of the parameter. Moreover, assuming a spatially homogeneous adhesion is a model simplification. In reality, the membrane-substrate adhesion is probably not spatially uniform, which could introduce additional perturbations to the wave propagation. An interesting example is a localized adhesion with an extreme strength that clamps membrane and hence diverts the propagation of the wave. This is both evidenced in our experiments and recapitulated by our model, lending further support for the mechanochemical nature of our wave. In the revision, we incorporated this new result as supplementary Fig. 14, and further clarified our simplified assumption on uniform membrane-substrate adhesion energy (line 451-456).

In the simulation a free boundary condition is imposed. In the experiments, as for example seen in Fig. 1(e), the area where the cell is close to the substrate is finite. In which way do the authors expect the boundary condition to affect the wave formation?

Response 3.2: We apologize for not being clear about the boundary condition. Our simulation is carried out on a membrane patch of 40 micron in diameter, wherein the membrane is clamped at the boundary. This boundary condition prevents membrane shape change and, hence, wave formation. This boundary condition recapitulates our experimental observations, which show that while waves propagate on the cell ventral side of the membrane, they disappear upon approaching the edge. In the updated draft, we further clarified the model description of the boundary condition and its implications in the wave formation at long times (line 133-135, 444-456, and the new supplementary Fig. 12).

The set of differential equations that the authors solve contains a term that couples the membrane shape to the concentration of the curved F-BAR proteins in Eq. (2.6), while a term that couples F-BAR motion within the lipid bilayer to the membrane curvature is missing in Eq. (2.3). Such a directed motion when the proteins follow the bilayer deformation can be expected to be much faster than diffusion of proteins in the bilayer. The authors have shown that diffusion can be neglected, but are there arguments that could indicate why the coupling of proteins to the membrane curvature occurs only via attachment of proteins from the cytosol (and not via curvature-driven motion of the proteins within the bilayer)?

Response 3.3: We thank the reviewer for bringing up this interesting idea. Indeed, the proteins could surf along the hydrodynamic flow of lipid within the bilayer, which is a real material propagation in space. How much this mode of motion contributes to the wave propagation in part depends on how long the proteins remain bound to the membrane. Experiments show that F-BAR turns over from membrane rapidly with a lifetime on the order of seconds (for instance, see Ramesh et al., *Sci Rep*, 2013), which is also consistent with our experimental observation that the $t_{1/2}$ of F-BAR puncta is a few seconds during the wave propagation (Fig. 5f). If without interference by others, single wave typically propagates ~ 20 seconds or longer in one round in our experiment (Fig. 5f). This separation in timescale suggests that this curvature-driven motion of the proteins within the bilayer, if exists, may not significantly contribute to wave propagation. We therefore assumed that this mode of protein motion was not significant. This is

consistent with our zoom-in kymograph (Fig. 5f), which shows that the F-BAR puncta does not move notably in the direction of the wave propagation. On the other hand, our model treated the membrane as an elastic sheet, which by its nature is incapable of describing the hydrodynamic in-plane flow of lipids within the bilayer. In our future study, we would like to systematically investigate how the in-plane lipid flow influences traveling wave in a more general setting. We further elaborated this model assumption in the revision (SI, pg. 6).

I would have expected that in immune cells the actin dynamics would also couple directly to the membrane shape dynamics and not only via an effective surface tension. However, this is not taken into account in Eqs. (2.1) to (2.6). Can the authors comment on this?

Response 3.4: We'd totally agree with the reviewer that the actin dynamics could directly influence the membrane shape (*i.e.*, the membrane height). By describing the membrane stiffening effect of actin polymerizations, our model only focuses on one of many possible effects mediated by actin dynamics. When we altered the model, which the actin polymerization directly increases the membrane-substrate adhesion energy, instead of membrane tension, we showed that the essential features of our conclusion remain robust, including 1) the coupling of membrane undulation with the protein traveling waves, and 2) the protein wave propagation reflects local protein assembly from cytoplasm, rather than lateral diffusion along the cortex (please see the new supplementary Fig. 11c, and its discussion in the revised main text, line 463-465).

Regarding the conclusions, a wave that travels faster than the speed of light cannot convey information. I am not sure whether this is what the authors write in the last paragraph of the main text? Also, I do not understand whether a chain of lamps flashing in quick succession should be an example for a wave that is faster than the speed of light? I recommend that the authors reconsider the wording in this paragraph.

Response 3.5: In the new draft, we removed this confusing part of discussion, as it is not essential to the conclusion.

Reviewers' comments:

Reviewer #1 (Remarks to the Author):

The authors have address my concerns in the previous review in a sufficient manner. I recommend publication

Reviewer #2 (Remarks to the Author):

This reviewer appreciates the effort the authors put into the re-writing Results and Discussion. This part has improved including both content and quality of English language (but see below). Unfortunately, fundamental requests raised upon first round of review regarding the behavior of the model have not been addressed. The authors were asked to demonstrated oscillatory regime of their model that matches the time series depicted in multiple panels of figures 1 and 2. Below is the point 5 of the original review:

"Presentation of modeling results – related to the above point 4. Figures 1 and 2 show extremely small domains of space on which the waves emerge from the center, presumably induced by the local perturbation shown in Fig. 1b, which argues against the oscillatory nature of the dynamics. The domain is so small that even one wavelength does not fit into it. The movie S2 does not help the situation, since the domain is the same. This makes the reader wonder if these waves are indeed stationary phenomenon or only transients to some other type of dynamics. The caption in Fig. 2 says "for the purpose of demonstration (?), the snapshots... only show a part of simulated membrane patch". This is clearly not true for the Fig. 2a and Movie S2 because we see that the membrane is clamped at the boundary of the shown domain. This is surprising and makes the reader wonder if the authors hide some dynamics which they don't want the reader to see. Furthermore, Fig. 2c clearly suggests that the dynamics is not periodic, the profile of waves continuously changes and the last wave-length is clearly distinct from the middle. Is this because the waves are not stationary?"

This request was entirely ignored by the authors. Instead their new Figures 1, 2, Supp. Figure 2 and two first movies show exactly what they used to show before, first 17 to 30 sec of the model dynamics after it was induces by the pulse shown in Figure 1b. It is clear from Figs. 1d, e, 2c that the period of their dynamics is about 25 sec. Therefore simulations presented as movies or stills shown in Fig. 2a should show long-term dynamics corresponding to at least 2-3 time periods, that is, 50-75 seconds. It is not clear what happens when the first wave induced by the pulse of Cdc42 activation (Fig. 1b) hits the boundary. Supp. Fig. 2 suggests that new maximum arising at the center of the system is going to have much smaller amplitude, what about second, third, etc.? What initiates the second, third (in fact any successive) maximum of dynamics in the middle of the circular domain? If the authors continue to pulse the system with 25 sec-periodic excitation identical to the pulse shown in Fig. 1b, it is absolutely fine. But in this case, they have to clearly acknowledge that their dynamics is excitable, that it is induced by periodic central excitation. It is not clear why in the system center, where the curvature of the membrane is negative and only increases in the amplitude (even more negative!) as the wave propagates to the boundary, a new maximum of height (and thus positive curvature) should arise on its own.

If the authors are not willing to provide this required information in the form of movies and figures, they have to provide their COMSOL model code with the parameters set to match the dynamics shown in Figures 1 and 2. Without this reproduction of their results by the reviewers, it is not clear if the dynamics shown in Figs. 1e (low panels), Fig. 2a, movies 1 and 2 matches the time series shown in Figs. 1d, 1e (upper panel), 2c. Until this reviewer is satisfied that the model behaves as it is claimed to behave by the authors, there is no point in discussing the authors conclusions any further.

English language. It is rewarding to see that Results and Discussion have been re-written by a more proficient English speaker. However, the Abstract, Introduction and figure captions (not to mention supplement) have remained untouched. In only first two pages (Abstract and Introduction) we see the same "diffusions", "evidences", "sensitivities" and "curvatures". This is embarrassing, please address English across the entire text of the paper and not only in the Results and Discussion.

Reviewer #3 (Remarks to the Author):

The authors have adequately dealt with my concerns.

Reviewer #2 (Remarks to the Author):

This reviewer appreciates the effort the authors put into the re-writing Results and Discussion. This part has improved including both content and quality of English language (but see below).

Unfortunately, fundamental requests raised upon first round of review regarding the behavior of the model have not been addressed. The authors were asked to demonstrate oscillatory regime of their model that matches the time series depicted in multiple panels of figures 1 and 2. Below is the point 5 of the original review: "Presentation of modeling results – related to the above point 4. Figures 1 and 2 show extremely small domains of space on which the waves emerge from the center, presumably induced by the local perturbation shown in Fig. 1b, which argues against the oscillatory nature of the dynamics. The domain is so small that even one wavelength does not fit into it. The movie S2 does not help the situation, since the domain is the same. This makes the reader wonder if these waves are indeed stationary phenomenon or only transients to some other type of dynamics. The caption in Fig. 2 says "for the purpose of demonstration (?), the snapshots... only show a part of simulated membrane patch". This is clearly not true for the Fig. 2a and Movie S2 because we see that the membrane is clamped at the boundary of the shown domain. This is surprising and makes the reader wonder if the authors hide some dynamics which they don't want the reader to see. Furthermore, Fig. 2c clearly suggests that the dynamics is not periodic, the profile of waves continuously changes and the last wavelength is clearly distinct from the middle. Is this because the waves are not stationary?"

This request was entirely ignored by the authors. Instead their new Figures 1, 2, Supp. Figure 2 and two first movies show exactly what they used to show before, first 17 to 30 sec of the model dynamics after it was induced by the pulse shown in Figure 1b. It is clear from Figs. 1d, e, 2c that the period of their dynamics is about 25 sec. Therefore simulations presented as movies or stills shown in Fig. 2a should show long-term dynamics corresponding to at least 2-3 time periods, that is, 50-75 seconds. It is not clear what happens when the first wave induced by the pulse of Cdc42 activation (Fig. 1b) hits the boundary. Supp. Fig. 2 suggests that new maximum arising at the center of the system is going to have much smaller amplitude, what about second, third, etc.? What initiates the second, third (in fact any successive) maximum of dynamics in the middle of the circular domain? If the authors continue to pulse the system with 25 sec-periodic excitation identical to the pulse shown in Fig. 1b, it is absolutely fine. But in this case, they have to clearly acknowledge that their dynamics is excitable, that it is induced by periodic central excitation. It is not clear why in the system center, where the curvature of the membrane is negative and only increases in the amplitude (even more negative!) as the wave propagates to the boundary, a new maximum of height (and thus positive curvature) should arise on its own.

If the authors are not willing to provide this required information in the form of movies and figures, they have to provide their COMSOL model code with the parameters set to match the dynamics shown in Figures 1 and 2. Without this reproduction of their results by the reviewers, it is not clear if the dynamics shown in Figs. 1e (low panels), Fig. 2a, movies 1 and 2 matches the time series shown in Figs. 1d, 1e (upper panel), 2c. Until this reviewer is satisfied that the model behaves as it is claimed to behave by the authors, there is no point in discussing the authors conclusions any further.

Response 2.1: We apologize that we misunderstood some part of the original comment #5 by this reviewer. We did not purposely ignore them. In this revision, we tried our best to address the requests.

First of all, in the model we do not “pulse the system with 25 sec-periodic excitation identical to the pulse shown in Fig. 1b”. The activation signal is localized (within a range of ~ 1 micron) and transient (only lasting ~ several seconds). We specifically clarified in the new revision that, after the 5-second time mark, there is no external activation signal anymore in our model (see the updated Fig. 1b and the caption).

Second, we now provided the new Supplemental Movie 2 and Supplemental Fig. 2, which show our model simulation results with a long-term dynamics of membrane shape changes coupled with cortical protein dynamics. We referred this point in the relevant discussion of Fig. 2a in the main text.

Third, in the new Supplemental Fig. 12a, we now provided the detailed simulation result showing that as the wave approaches the boundary, the clamped boundary condition clamps the membrane back to the baseline. This flattening of membrane disrupts the wave. This is consistent with our experimental observation that the wave disappears as it hits the cell ventral boundary (see Supplemental Movies 3-5).

Finally, after the first round of wave propagates outward, new wave initiates at the original epicenter. The reason is that after the first F-BAR wave has passed, the actin level returns to baseline at the epicenter. Because actin level modulates cortex tension, cortex tension is much lower at the epicenter than in the periphery. Accordingly, membrane shape relaxes more slowly, resulting in a “residual” membrane bump. This is indicated by the 4th panel in the new Supplemental Fig. 2b (the brighter cyan color at the middle at 30 second). It is this residual membrane shape deformation that serves as a cue to recruit F-BAR, which in turn initiates the next round of oscillation. Furthermore, because the clamped boundary dampens the membrane shape changes, its effect will propagate from the edge to the epicenter, where it flattens the residual membrane shape deformation over time. Without external activation signal, this flattening of membrane eventually prevents the new round of F-BAR cortical recruitment and, hence, dampens the oscillation and wave after all. We further clarified these points in the new draft (pg. 12) and in new Supplemental Fig. 2 along with the figure caption.

English language. It is rewarding to see that Results and Discussion have been re-written by a more proficient English speaker. However, the Abstract, Introduction and figure captions (not to mention supplement) have remained untouched. In only first two pages (Abstract and Introduction) we see the same “diffusions”, “evidences”, “sensitivities” and “curvatures”. This is embarrassing, please address English across the entire text of the paper and not only in the Results and Discussion.

Response 2.2: We further streamlined the language of the entire paper.

REVIEWERS' COMMENTS:

Reviewer #2 (Remarks to the Author):

The manuscript by Wu et al. has been substantially revamped and clearly benefited from a major rewriting. Both the clarity of presentation and the quality of English have been significantly improved to merit the publication. The authors have gone a long way since the original version of the manuscript.

This reviewer is satisfied with the changes and the specific information provided about the behavior of the model. The authors have eventually explained the regime of oscillation initiation and their transient nature. Given the complexity of the mechano-chemical model, it is possible that its behavior will change on a larger spatial domain, something that the modelers may want to explore in their future work. However, the information provided in the present manuscript is sufficient to illustrate the qualitative message of the paper.